# Dysregulated transforming growth factor-beta mediates early bone marrow dysfunction in diabetes

Jina J. Y. Kum[1], Christopher J. Howlett[1,2,3 ✉] & Zia A. Khan [1,3,4 ✉]

Diabetes affects select organs such as the eyes, kidney, heart, and brain. Our recent studies show that diabetes also enhances adipogenesis in the bone marrow and reduces the number of marrow-resident vascular regenerative stem cells. In the current study, we have performed a detailed spatio-temporal examination to identify the early changes that are induced by diabetes in the bone marrow. Here we show that short-term diabetes causes structural and molecular changes in the marrow, including enhanced adipogenesis in tibiae of mice, prior to stem cell depletion. This enhanced adipogenesis was associated with suppressed transforming growth factor-beta (TGFB) signaling. Using human bone marrow-derived mesenchymal progenitor cells, we show that TGFB pathway suppresses adipogenic differentiation through TGFB-activated kinase 1 (TAK1). These findings may inform the development of novel therapeutic targets for patients with diabetes to restore regenerative stem cell function.

[1] Pathology and Laboratory Medicine, Schulich School of Medicine & Dentistry, Western University, London, ON, Canada. [2] Pathology and Laboratory Medicine, London Health Sciences Centre, London, ON, Canada. [3] Lawson Research Institute, London, ON, Canada. [4] Division of Genetics & Development, Children's Health Research Institute, London, ON, Canada. ✉email: christopher.howlett@lhsc.on.ca; zia.khan@schulich.uwo.ca

The global prevalence of diabetes has been steadily increasing worldwide and is projected to affect 10.4% of all adults by 2040[1]. Diabetes poses a substantial burden on the health care system due to the challenges associated with managing various secondary complications. These complications include retinopathy, cardiomyopathy, nephropathy, and atherosclerosis. More recently, human diabetes and experimental models of the disease have shown that the bone marrow may also be a target organ of diabetic complications[2–4]. Both enhanced bone marrow adiposity and skeletal fragility have been well documented[5–7]. Diabetes has been shown to cause numerous other abnormalities[8] in the bone marrow, including small vessel rarefaction[5], reduced hematopoietic stem cell retention and quiescence factor expression[9,10], myelopoiesis[9,11], and neuropathy[12]. Some of these abnormalities, such as enhanced myelopoiesis/increased monocyte generation[13], blunted hematopoietic stem cell mobilization[14], and lower bone marrow-derived endothelial colony forming cells[15–18] may increase the risk of cardiovascular disease in diabetes. Although the pathogenesis of these bone marrow abnormalities is slowly being uncovered, evidence exists that suggests enhanced bone marrow adipogenesis leads to hematopoietic and non-hematopoietic stem cell deficits. For example, we and others have observed depletion of non-hematopoietic bone marrow stem cells coupled to enhanced marrow adiposity[6,19]. In addition, examination of rodent bones with varying levels of adipocytes, and bones from A-ZIP/F1 fatless mice show that adipocytes in the bone marrow negatively regulate hematopoietic stem cells[20]. Importantly, A-ZIP/F1 mice—which show polyuria, polydipsia, and hyperglycemia[21]—illustrate that it is not simply the diabetic milieu but the presence of adipocytes that reduces hematopoietic stem cells in bones. Collectively, depletion of bone marrow regenerative cells, likely driven by enhanced adipogenesis, may underlie the development of other secondary vascular complications of diabetes. Therefore, it is imperative that we understand how diabetes enhances adipogenesis and leads to the depletion of marrow-resident stem cells.

Identifying mechanisms that cause enhanced bone marrow adipogenesis in diabetes and when these alterations take place, in relation to other complications, requires an experimental model that recapitulates the human disease. Streptozotocin (STZ)-induced diabetic mice and rats are the most established and recommended by the Animal Models of Diabetic Complications Consortium (http://www.diacomp.org). Using this model, researchers have produced a precise timeline of pathological changes in target organs. For example, for diabetic retinopathy, retinal ganglion cells are reduced at 6 weeks[22,23], and capillary basement membrane thickening is evident at 17 weeks of hyperglycemia[24]. Similarly, for diabetic nephropathy, glomerular hypertrophy, glomerular basement membrane thickening, and mesangial expansion take more than 6 months of diabetes[25,26]. For cardiac dysfunction, increased left ventricle fibrosis is evident at 8 weeks following STZ-induced diabetes[27]. Collectively, these studies provide a benchmark to identify bone marrow dysfunction in diabetes.

Since bone marrow adipocytes are considered negative regulators of marrow-resident stem cells, we hypothesized that enhanced marrow adiposity in diabetes will precede stem cell depletion. To test our hypothesis, we examined the effect of STZ-induced diabetes in bone tissues of mice. Our results show that 1 month of diabetes is sufficient to enhance marrow adiposity in tibiae of mice. This alteration was observed before deficits in bone hematopoietic fraction and the depletion of bone marrow-resident stem cells. Additionally, we show that STZ-induced hyperglycemia suppresses transforming growth factor-beta (TGFB) pathway in the bone marrow. In cultured primary bone marrow cells, we demonstrate that high levels of glucose suppress TGFB, and the addition of exogenous TGFB inhibits adipogenesis through the non-canonical TGFB-activated kinase 1 (TAK1) pathway. Our studies have provided fundamental insight into diabetic marrow disease and identified the potential underlying mechanisms.

## Results

**Staging of streptozotocin-induced diabetic mice to detect early dysfunction in organs**. With the goal of identifying early changes in the bone marrow of diabetic mice, we induced diabetes in 5-week-old C57BL/6 mice (corresponding to 15–20 human years). We selected this age to identify diabetes-specific adipogenesis. We know that bone marrow adiposity increases with age[28]. In C57BL/6 mice, age-related bone marrow adiposity is seen from 12 weeks to 56 weeks of age[28]. Therefore, we induced diabetes in 5-week-old mice with multiple low doses of streptozotocin (STZ) on 5 consecutive days (Fig. 1a). STZ-injected mice showed significantly lower body weights and higher blood glucose levels, 1 week after the last STZ injection (d0; Supplementary Fig. S1a, b) and after 1 month (Supplementary Fig. S1c, d). As expected, the cumulative body weight difference in the STZ-induced diabetic mice was significantly less compared with the control mice (Supplementary Fig. S1e). Histological analysis of H&E and insulin antibody-stained pancreata displayed significantly reduced size of the islets, and the number of insulin-immunoreactive areas in STZ-induced mice compared with the control mice (Supplementary Fig. S2).

We know that, histologically, diabetic complications manifest as increased extracellular matrix deposition and basement membrane thickening[29], cell loss[30], infiltration of myeloid cells and inflammation in target organs[31]. These structural features become obvious in STZ-induced models at different times, but typically require 2-plus months of diabetes duration. Examination of H&E-stained sections of the heart, liver, kidney, and lung did not reveal any significant changes in STZ-induced diabetic mice compared to the non-diabetic mice at 1 month of disease duration (Supplementary Fig. S3a). Retina, heart, and kidney sections stained with Periodic Acid-Schiff (PAS) to detect changes in polysaccharide deposition also showed no alterations (Supplementary Fig. S3b). Next, we stained the tissues with Picro-Sirius Red to highlight collagen deposition and observed no significant fibrosis in the tissues (Supplementary Fig. S3c–f). These results show that 1 month of hyperglycemia is not sufficient to induce hallmark structural changes in well-known target organs. Analysis of retina tissues revealed a significant decrease in the thickness of the inner nuclear layer in the diabetic mice (Supplementary Fig. S4), whereas the other retinal layers did not change. This observation is consistent with a recent study in diabetic rats highlighting early structural changes in the retina at 1 month[32].

Epididymal fat of mice showed a difference between the STZ-induced diabetic mice and the non-diabetic controls. Diabetes was associated with increased nuclei per tissue area (Supplementary Fig. S5a–c) and suppressed levels of adipogenesis-associated genes, including *Pparg*, *Cepba*, *Fabp4*, *Lpl*, and *Adipoq* (Supplementary Fig. S5d–h). Increased lipolysis[33,34] in this model of type 1 diabetes may explain suppressed adipogenesis-related genes and increased nuclei.

Detailed examination of liver tissues harvested from the STZ-diabetic mice showed a marked increase in vacuole accumulation, as highlighted by PAS-diastase staining (Supplementary Fig. S6a). To determine whether these vacuoles are lipid droplets, we stained the liver sections with perilipin-1 (PLIN1), a selective lipid droplet-associated protein[35]. Interestingly, liver tissues isolated from STZ-induced diabetic mice displayed robust PLIN1

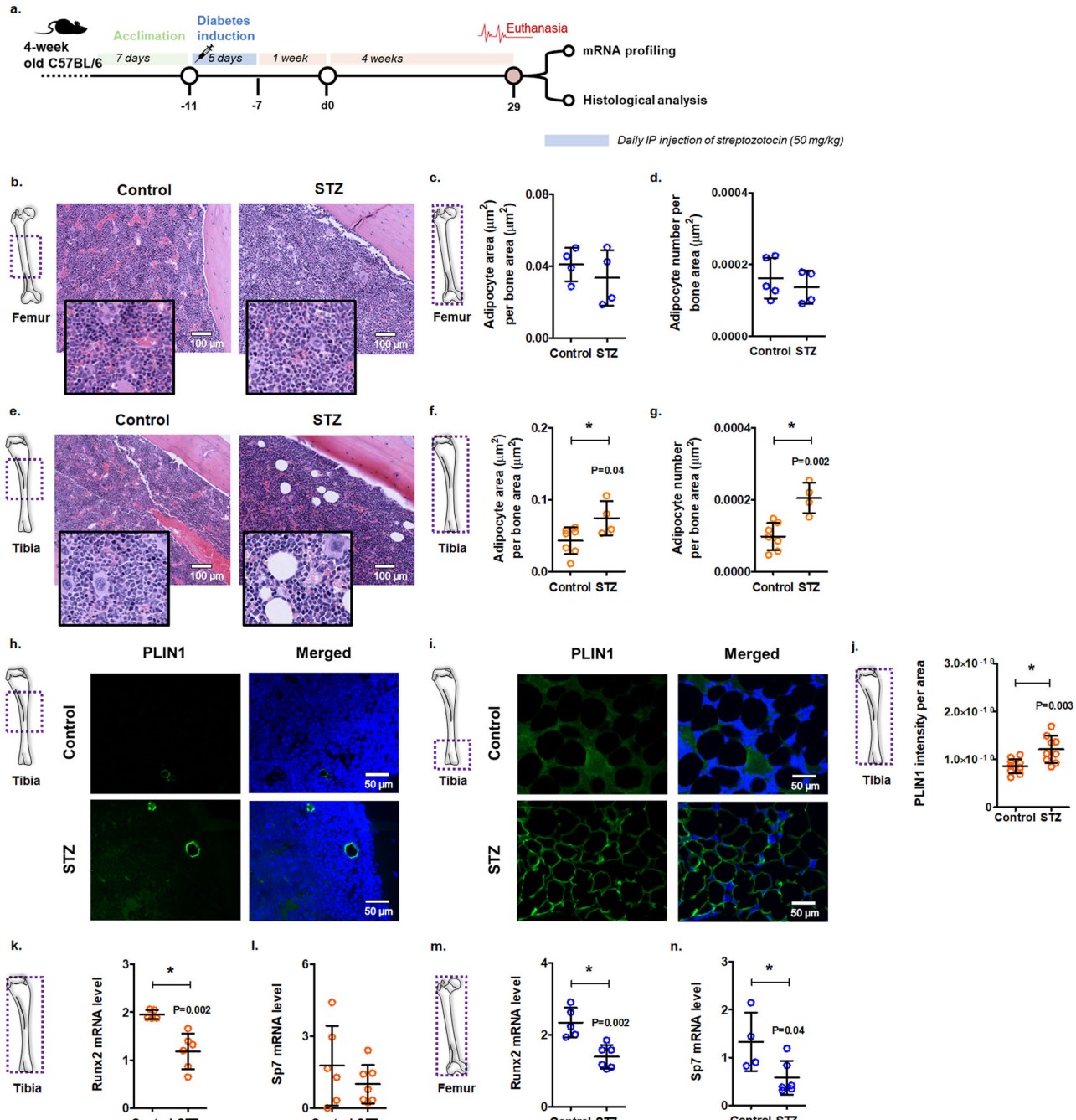

**Fig. 1 Streptozotocin-induced diabetes enhances adipocyte number and area in tibiae of mice at 1 month. a** Experimental scheme for diabetic mouse study. Diabetes was induced in C57BL/6 mice with daily intraperitoneal injections of streptozotocin (STZ; 50 mg/kg) for 5 consecutive days. Non-diabetic control mice received an equal volume of citrate buffer. Blood glucose levels were checked 1 week after the last STZ injection to confirm hyperglycemia (d0). **b** Representative H&E-stained images of the mouse femur [scale bar = 100 μm]. Inserts showing higher magnification. Quantification of adipocyte area per bone area (**c**) and adipocyte number per bone area (**d**) in the femurs. Parameters were measured using *MarrowQuant* [Mean ± SD; $n = 4$ (5 for control in panel **d**); each data point represents a mouse; two-tailed student's *t*-test: *$p < 0.05$]. **e** Representative images of the H&E-stained mouse tibia [scale bar = 100 μm]. Inserts showing higher magnification. Quantification of adipocyte area per bone area (**f**) and adipocyte number per bone area (**g**) in the tibiae [Mean ± SD; $n = 7$ control and 4 STZ in panels **f** and **g**; each data point represents a mouse; two-tailed student's *t*-test: *$p < 0.05$]. Immunofluorescence staining of mouse tibia for perilipin-1 (PLIN1; green). Sections were counterstained with DAPI (blue) [scale bar = 50 μm]. Figure showing shaft (**h**) and distal regions (**i**) of the tibia. **j** Quantification of PLIN1 intensity per area, as determined by *ImageJ* [Mean ± SD; $n = 10$ control and 9 STZ; two-tailed student's *t*-test: *$p < 0.05$]. mRNA levels of osteogenesis-associated genes were measured in the tibiae (**k, l**) and femurs (**m, n**) of control and diabetic (STZ) mice [Data normalized to *Actb* and *Gapdh*; Mean ± SD; $n = 5$ control and 6 STZ in panel **k**, 4 control and 6 STZ in panel **l**, 5 control and 6 STZ in panel **m**; each data point represents a mouse; two-tailed student's *t*-test: *$p < 0.05$].

immunoreactivity indicating that the vacuoles are indeed lipid droplets (Supplementary Fig. S6b). Collectively, these results suggest that the STZ-induced model of diabetes is effective and causes only early changes in mice after 1 month.

**STZ-induced diabetic mice show early enhanced adipogenesis in the tibiae.** Tibiae and femurs were isolated for the discovery of early pathogenetic changes in the bone tissues that may entail disrupted osteo- and adipogenesis. No changes in the lengths of the bones were observed between the two experimental groups (Supplementary Fig. S7). Using *MarrowQuant*, a semi-automated image analysis plug-in for *QuPath*[36], we observed a significantly increased area and number of adipocytes in the tibia of diabetic mice but not in the femur (Figs. 1b–g). These findings suggest that diabetes, at least early on, specifically disrupts the tibia. To confirm this finding, we stained the tibia sections with PLIN1 and observed a significant increase in PLIN1 intensity in the marrows of diabetic mice (Figs. 1h–j). In the control mice, PLIN1-positive adipocytes were only found in the distal tibia (Fig. 1i). It is known that distal tibia contains constitutive adipose tissue that develops soon after birth and is relatively stable. Regulated adipose, on the other hand, is found in the shaft and proximal regions of tibia and responds to a wide range of changes including nutritional, physiological, and genetic[37]. Consistent with this classification, we found diabetes to enhance regulated adipose in shaft regions of the tibia. To supplement these results, we prepared a marrow flush from tibiae and femurs of the mice (Supplementary Fig. S8) to measure gene expression levels. We first examined adipogenesis-associated genes and show that transcriptional factors *Cebpa* and *Pparg*, and adipocyte-related adiponectin (*Adipoq*) are not increased in the tibia (nor femur) after 1 month of diabetes (Supplementary Fig. S9). However, analysis of osteogenesis-related genes showed a significant reduction in *Runx2* in both the tibia and femur of the diabetic mice (Figs. 1k–n). Another osteoblast transcription factor, *Sp7*, was only downregulated in the femur but not the tibia at 1 month. Previous studies have shown that RUNX2 is the first transcription factor required for determination of the osteoblast lineage, while SP7 further directs the fate of mesenchymal cells to osteoblasts[38]. These findings show that diabetes may cause suppressed osteoblastogenesis in mice, and the effects may be more pronounced in the femur compared to the tibia.

Bone marrow adipogenesis in an STZ-induced type 1 model of diabetes in our current study and other previous studies[39–41] suggests that the process is insulin-independent. However, insulin may enhance the process through the regulation of fatty acid transporters[42,43] and induction of *Pparg*[44]. Acyl-CoA synthetase long chain family member 1[45] (ACSL1), CD36[46], and solute carrier family 27 member 1[43] (also known as fatty acid transport protein 1, FATP1) have been shown to mediate lipid accumulation in various cell types. We first measured the expression of these fatty acid transporters in human bone marrow-derived mesenchymal progenitor cells (bm-MPCs; precursors of adipocytes) and show that the three fatty acid transporters are expressed robustly (housekeeping *ACTB*, threshold cycle (CT) 28.4 ± 0.15 (Mean ± SD); *ACSL1*, CT 28.31 ± 0.02; *CD36*, CT 30.19 ± 0.05; *SLC27A1*, CT 30.47 ± 0.06). We then tested whether bm-MPCs are capable of adipogenic differentiation in an insulin-independent manner. We produced an insulin-deficient adipogenesis induction media (idADP) containing serum, dexamethasone, isobutylmethylxanthine, and indomethacin (Supplementary Table S1). We then supplemented this base idADP with various test agents including high glucose (HG; 25 mmol/L), linoleic acid-oleic acid (LIN/OL; a source of fatty acids), or a combination. Lipid-laden cells were seen in idADP and were increased in the

presence of HG and/or LIN/OL (Supplementary Fig. S10). We know that STZ induction of hyperglycemia in mice increases triglycerides and free fatty acids as early as 2 weeks following onset[47]. Therefore, our results suggest that adipogenesis in the bone marrow is insulin-independent and is enhanced by hyperglycemia and the availability of fatty acids.

Enhanced adipogenesis in the tibia of STZ-induced diabetic mice at 1 month of diabetes did not appear to be associated with any alteration in the marrow cellularity/nucleated area (majority of the cells being of hematopoietic origin) and *Cd45* (*Ptprc*) mRNA levels (Supplementary Fig. S11). We and others have previously shown that diabetes does not alter the number of CD45-positive hematopoietic cells in experimental diabetes[6]. Other researchers have also shown that it takes 27–30 weeks of diabetes onset in rodents to decrease bone marrow cellularity[5]. In human type 2 diabetic patients, no changes in CD45-positive cells were noted[6,19]. Therefore, our results confirm previous reports of unaltered CD45-positive hematopoietic fraction at this early time point.

Since we and others have shown enhanced bone marrow adipogenesis to be associated with a reduction in non-hematopoietic stem cells in human diabetes[6,19], we stained tibiae of mice with stem cell antigen-1 (SCA1) and sex determining region Y-box 2 (SOX2). Functional differentiation assays and single-cell RNA sequencing have confirmed the identity of non-hematopoietic marrow stem cells as SCA1-positive[48–50]. Our data shows no changes in SCA1 or SOX2 immunoreactive cells (Fig. 2a–d). Since SCA1 has been reported on different marrow cell subpopulations including stromal cells[51], we further measured transcript levels of other stem cell genes *Sca1* (*Ly6a*), *Sox2*, *Oct4* (*Pou5f1*), and *Nanog*. Our results show no significant alteration in stem cell markers and transcripts in the bones of diabetic mice (Fig. 2e–l). These results show that bone marrow adiposity precedes tissue-resident stem cell depletion.

**Enhanced bone marrow adiposity in diabetes may be mediated through early disruption of the TGFB signaling pathway.** Very recently, TGFB pathway was shown to regulate lineage specification of mesenchymal progenitor cells in the marrow during development[52]. Therefore, we examined the influence of hyperglycemia on TGFB signaling in diabetic mice. In the tibia, *Tgfbr1*, *Tgfb1*, *Smad3*, *Smad6*, and *Bmp4* levels were significantly lower in STZ-induced diabetic mice compared with the non-diabetic controls (Fig. 3a–e). Similarly, in the femur, diminished levels of *Smad2*, *Smad3*, *Smad6*, and *Bmp4* were observed in the diabetic mice (Fig. 3f–i). Immunostaining of tibiae for TGFB1 largely confirmed gene expression results (Fig. 3j, k). Suppressed expression of the TGFB pathway genes in bone tissues of mice suggests that the TGFB pathway may be involved in early diabetes-induced changes in the bone tissues.

In an attempt to identify which cell types express TGFB1 and downregulate its expression in diabetes, we carried out in situ hybridization (ISH) in the bone tissues of mice. Previous studies have shown that TGFB1 is detectable in bone marrow cells, chondrocytes, and cartilaginous matrix in both neonatal and adult mice[53]. Our *Tgfb1* ISH showed a robust signal in megakaryocytes (Supplementary Fig. S12). This finding is consistent with a recent study showing that megakaryocytes express higher levels of *Tgfb1* compared to stromal cells[54]. Other bone marrow cells also expressed *Tgfb1*, but the signal was obscured by the compact morphology of the marrow cells and hematoxylin stain. Examination of these stained sections using mercury arc illumination showed that most bone marrow cells express *Tgfb1* and the likely reduction in *Tgfb1* in diabetic mice is attributable to most cells, not specific cell subsets (Supplementary

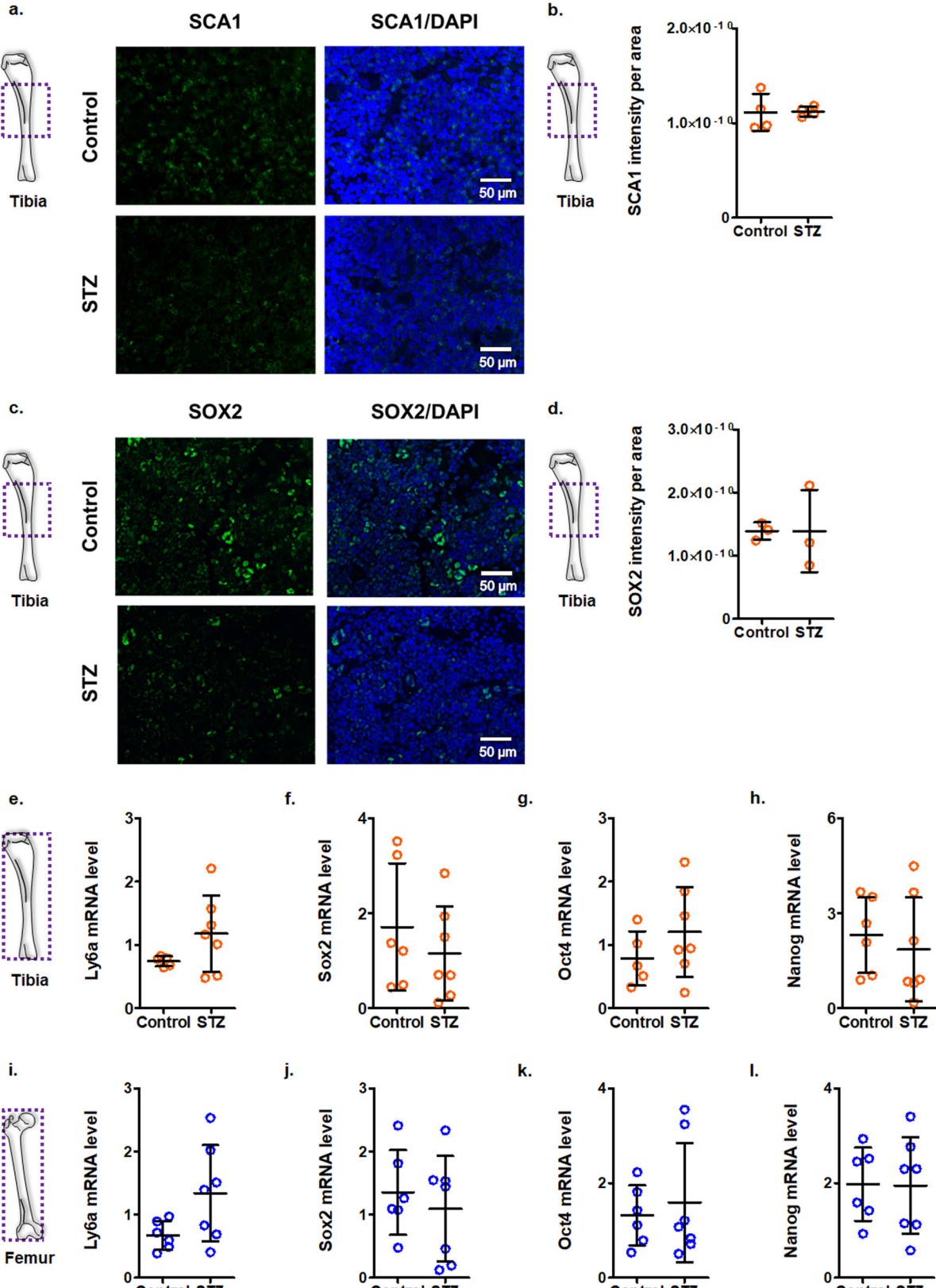

Fig. S13). Indeed, our immunohistochemical staining of the mouse tibia shows widespread presence of TGFB1 protein and overall reduction around all marrow cells (Fig. 3j).

We next examined bone tissues of longer-duration diabetes to determine whether TGFB suppression is maintained in diabetes and whether this associates with enhanced tissue adipogenesis. For

this, we analyzed the bone marrows of mice at 2 months of diabetes duration. As expected, diabetic mice weighed significantly less and had higher blood glucose levels compared with the non-diabetic control mice at 2 months (Supplementary Fig. S14). Other structural changes such as an increase in collagen deposition also appear in organs (Supplementary Fig. S15). Moreover, we observed

**Fig. 2 Stem cell antigens in the tibia of streptozotocin-induced diabetic mice at 1 month do not show any changes. a** Immunofluorescence staining of the tibiae of control and diabetic (STZ; 1 month) mice for SCA1 (green). Sections were counterstained with DAPI (blue) [scale bar = 50 μm]. **b** Quantification of SCA1 intensity per area, as determined by *ImageJ* [Mean ± SD; n = 4; two-tailed student's *t*-test: *p < 0.05]. **c** Immunofluorescence staining of the tibiae of control and diabetic (STZ; 1 month) mice for SOX2 (green). Sections were counterstained with DAPI (blue) [scale bar = 50 μm]. **d** Quantification of SOX2 intensity per area, as determined by *ImageJ* [Mean ± SD; n = 3; two-tailed student's *t*-test: *p < 0.05]. **e–h** mRNA levels of stem cell markers in tibia flush samples, showing *Sca1* (*Ly6a*), *Sox2*, *Oct4* (*Pou5f1*), and *Nanog* [Data normalized to *Actb* and *Gapdh*; Mean ± SD; n = 5 control and 7 STZ in panel **e**, 6 control and 7 STZ in panel **f**, 5 control and 7 STZ in panel **g**, 6 control and 7 STZ in panel **h**; each data point represents a mouse; two-tailed student's *t*-test: *p < 0.05]. **i–l** mRNA levels of stem cell markers in femur flush samples, showing *Sca1* (*Ly6a*), *Sox2*, *Oct4* (*Pou5f1*), and *Nanog* [Data normalized to *Actb* and *Gapdh*; Mean ± SD; n = 6 control and 7 STZ; each data point represents a mouse; two-tailed student's *t*-test: *p < 0.05].

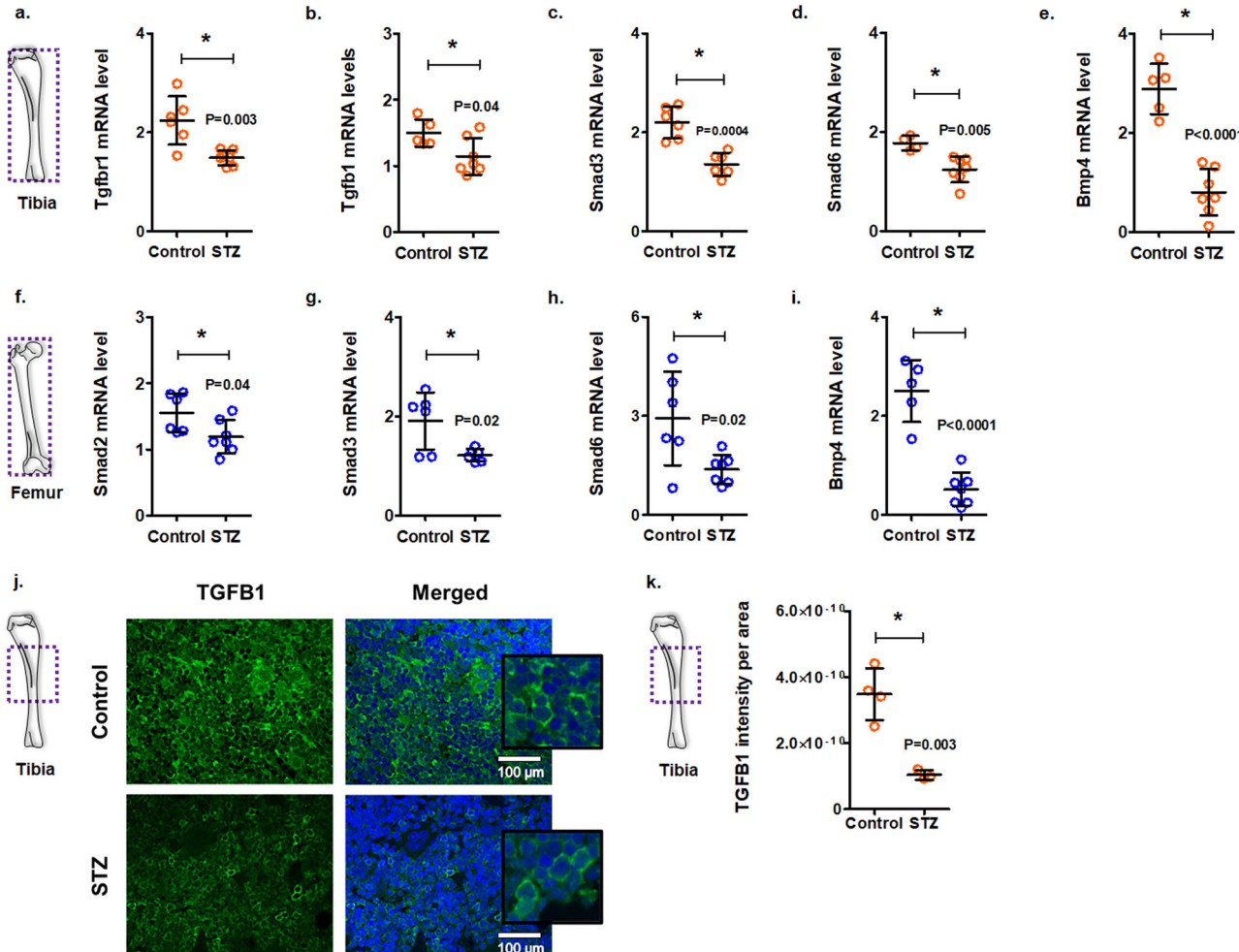

**Fig. 3 Suppressed TGFB pathway in the bone marrow of streptozotocin-induced diabetic mice at 1 month.** Tibia and femur samples were harvested from control and streptozotocin (STZ)-induced diabetic mice after 1 month. **a–e** mRNA levels of the TGFB pathway genes in the tibiae of mice [For panels **a**, **b**, data normalized to *Actb*, *Atp5f1*, and *Pgk1*; for panels **c–e**, data normalized to *Actb* and *Gapdh*; Mean ± SD; n = 6 control and 7 STZ in panel **a**, 5 control and 7 STZ in panel **b**, 6 control and 6 STZ in panel **c**, 4 control and 7 STZ in panel **d**, 5 control and 7 STZ in panel **e**; each data point represents a mouse; two-tailed student's *t*-test: *p < 0.05]. **f–i** mRNA levels of the TGFB pathway genes in the femurs of mice [For panels **f–i**, data normalized to *Actb* and *Gapdh*; Mean ± SD; n = 6 control and 7 STZ in panel **f**, 6 control and 6 STZ in panel **g**, 6 control and 7 STZ in panel **h**; 5 control and 7 STZ in panel **i**; each data point represents a mouse; two-tailed student's *t*-test: *p < 0.05]. **j** Immunostaining of tibia marrow for TGFB1 (green). Sections were counterstained with DAPI (blue) [scale bar = 100 μm]. Inserts showing higher magnification. **k** Quantification of TGFB1 intensity per area, as determined by *ImageJ* [Mean ± SD; n = 4 control and 3 STZ; two-tailed student's *t*-test: *p < 0.05].

enhanced adiposity in the tibia of diabetic mice, coupled with a significant increase in *Cebpa* mRNA (Fig. 4a–c). Similar to the 1-month diabetic mouse model, the femur showed no significant increase in adiposity nor upregulated levels of genes associated with adipogenesis even after 2 months (Fig. 4i–k). Interestingly, *Sp7* was downregulated in the tibia of diabetic mice at 2 months (Fig. 4e), unlike the results at 1 month. Furthermore, *Adipoq*, an adipocyte-related protein, and insulin-sensitizing hormone[55], levels were

significantly reduced in the bone marrow of diabetic mice (Fig. 4d, l). Importantly, analysis of bone tissues from diabetic mice at 2 months revealed a significant reduction in TGFB signaling in the tibia and femur (Fig. 4g–h and 4n–p), suggesting that these changes are sustained over time. Collectively, these findings show that hyperglycemia downregulates the TGFB signaling pathway in the bone marrow, which may contribute to the disruption of the marrow cellular composition.

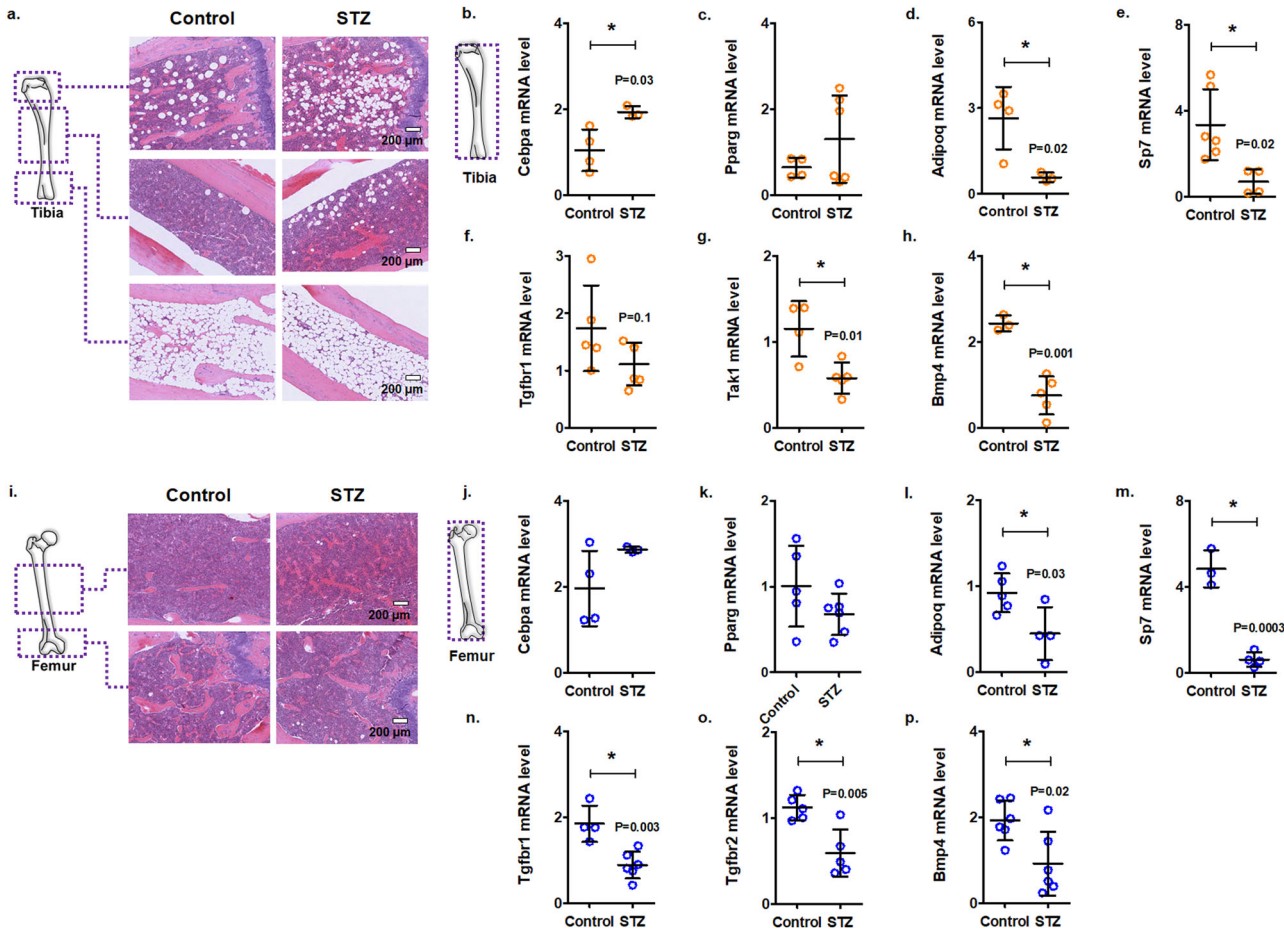

**Fig. 4 Enhanced adiposity in tibiae of mice after 2 months of streptozotocin-induced diabetes.** C57BL/6 male mice received streptozotocin (STZ; 50 mg/kg) or citrate buffer (non-diabetic controls). Tibiae were harvested 2 months after the onset of diabetes. **a** Representative H&E-stained sections of the tibia showing proximal, shaft, and distal regions [scale bar = 200 μm]. mRNA levels of adipogenesis-associated (**b**–**d**) and osteogenesis-associated (**e**) genes in marrow flush samples of mouse tibia [Data normalized to *Actb*, *Atp5f1*, and *Pgk1*; Mean ± SD; n = 4 control and 3 STZ in panel **b**, 4 control and 6 STZ in panel **c**, 4 control and 3 STZ in panel **d**, 6 control and 4 STZ in panel **e**; each data point represents a mouse; two-tailed student's *t*-test: *p < 0.05]. mRNA levels of TGFB pathway genes in the tibiae of control or diabetic (STZ) mice after 2 months of diabetes onset [Data normalized to *Actb* and *Gapdh*; Mean ± SD; n = 4 control and 5 STZ in panels **f** and **g**, 3 control and 5 STZ in panel **h**; each data point represents a mouse; two-tailed student's *t*-test: *p < 0.05]. **i** Representative H&E-stained sections of the femur showing shaft and distal regions [scale bar = 200 μm]. mRNA levels of adipogenesis-associated (**j**–**l**) and osteogenesis-associated (**m**) genes in marrow flush samples of mouse femur [Data normalized to *Actb*, *Atp5f1*, and *Pgk1*; Mean ± SD; n = 4 control and 3 STZ in panel **j**, 5 control and 6 STZ in panel **k**, 5 control and 4 STZ in panel **l**, 3 control and 4 STZ in panel **m**; each data point represents a mouse; two-tailed student's *t*-test: *p < 0.05]. mRNA levels of TGFB pathway genes in the femurs of control or diabetic (STZ) mice after 2 months of diabetes onset [Data normalized to *Actb* and *Gapdh*; Mean ± SD; n = 4 control and 6 STZ in panel **n**, 5 control and 5 STZ in panel **o**, 6 control and 6 STZ in panel **p**; each data point represents a mouse; two-tailed student's *t*-test: *p < 0.05].

A recent study found that the deletion of *Tgfbr2* in *Osx-Cre-targeted* mesenchymal cells led to changes in *Cxcl12* mRNA expression in the bone marrow[52]. Since our studies identified suppressed TGFB signaling in diabetes, we examined if stem cell niche factors, including CXCL12, were affected. As anticipated, there was a significant reduction in *Cxcl12* levels in the tibia and femur of diabetic mice (Supplementary Fig. S16a, e). Unlike *Cxcl12*, however, no significant changes were seen in *Kit* and *Kitl* levels in the tibia or femur (Supplementary Fig. S16c, d, g, h), indicating specificity in the changes. Furthermore, levels of other known niche factors *Angpt1*, *Icam1*, and *Vcam1* were not significantly altered in the bone marrow of diabetic mice (Supplementary Fig. S17). In support of a selective effect on *Cxcl12* is the observation that the CXCL12 receptor, *Cxcr4*, which is expressed on stem cells was not altered in the tibia nor femur (Supplementary Fig. S16b, f). We also found that the CXCL12 staining intensity in the bone marrow was reduced in bone tissues of diabetic mice (Supplementary Fig. S16i, j).

Analysis of tissues from mice at 2 months of diabetes duration revealed a sustained reduction in *Cxcl12* expression in the tibia and femur (Supplementary Fig. S16k, l), suggesting that these changes are also sustained over time, much like TGFB suppression. Such changes did not appear to be mediated through a reduction in CXCL12-expressing cells[56,57] (known as CAR cells), as leptin receptor (LEPR), a marker of CAR cells[58,59], did not show significant alteration in the tibia of 1-month diabetic mice (Supplementary Fig. S18).

**Wnt signaling pathway is not altered in bone marrows of diabetic mice at 1-month disease duration.** Our laboratory has previously found that non-canonical WNT11 mediates high glucose-induced adipogenic differentiation of MPCs in culture[6]. Additionally, previous studies have reported a potential crosstalk between the TGFB and Wnt signaling pathways[60]. Therefore, we examined whether canonical and non-canonical Wnt signaling

pathways may be altered in STZ-induced model of early hyperglycemic alterations. We used *Ctnnb1* and *Wnt11* levels as a proxy for the canonical and non-canonical Wnt signaling, respectively. Analysis of bone tissues, however, showed no changes in the Wnt signaling pathway (Supplementary Fig. S19). These findings suggest that TGFB alteration may precede Wnt alteration in diabetes-induced bone marrow deficits.

**High glucose exposure affects TGFB1 signaling pathway in cultured bone marrow mesenchymal progenitor cells**. We have previously shown that culture conditions that mimic diabetes cause enhanced adipogenic differentiation in human bm-MPCs[6]. Furthermore, bm-MPCs from type 2 diabetic patients express high levels of *Cebpa*, favouring adipogenic differentiation[61]. Therefore, we used bm-MPCs to examine the signaling mechanisms in detail. We first investigated the effect of high levels of glucose on the TGFB pathway by culturing cells in normal growth media containing elevated levels of glucose for 21 days (Supplementary Fig. S20). We screened for genes associated with adipogenic differentiation and show that cells cultured in media containing 25 mmol/L glucose (high glucose; HG) exhibited suppressed mRNA levels of the TGFB family, including the downstream signaling mediators, *SMAD2*, *SMAD3*, and *SMAD4*, compared with cells cultured in media with 5 mmol/L glucose (Supplementary Fig. S20). In contrast to the TGFB pathway, no changes were observed in the fibroblast growth factor and insulin-like growth factor signaling pathways, other known adipogenesis-inducing pathways. This suggests that hyperglycemia potentially suppresses TGFB signaling in bm-MPCs to regulate cell fate changes.

We next explored the effect of exogenously supplied TGFB1 on adipogenic differentiation of bm-MPCs. We exposed bm-MPCs to adipogenic differentiation media with or without 10 ng/mL TGFB1 for 72 h. This time point has been previously reported to be sufficient for cultured cells to initiate differentiation[62,63]. TGFB1 exposure inhibited adipogenic differentiation of cells as illustrated by a significant reduction in *PPARG2* (Fig. 5a), which supports the findings from a mouse pre-adipocyte cell line[64]. Moreover, TGFB1 reduced the number of adipocytes in the culture and the level of lipid accumulation in the cells (Figs. 5b–d). We next tested whether elevated glucose levels further increase adipogenic programming. The addition of 25 mmol/L glucose to adipogenic differentiation media significantly increased *PPARG2* levels compared with adipogenic differentiation media with only 5 mmol/L glucose (Figs. 5e, f). These results are in alignment with our previous studies[65] and support the hypothesis that hyperglycemia in diabetes enhances bm-MPC adipogenesis. Interestingly, the addition of exogenous TGFB1 in high-glucose adipogenic media decreased *PPARG2* to levels comparable to adipogenic media with 5 mmol/L glucose, and reduced lipid content within the cells (Figs. 5e, f).

**Noncanonical TGFB signaling inhibits adipogenic differentiation of bone marrow mesenchymal progenitor cells**. Next, we used a variety of inhibitors of the TGFB signaling pathway to identify which arm of the TGFB pathway may be inhibiting adipogenesis in bm-MPCs. The inhibition of ALK5, TGFB-associated kinase 1 (TAK1), and JNK significantly rescued *PPARG2* and lipid accumulation in cells, negating the effects of TGFB1 (Figs. 6a, b). This triggered the question of whether cells would enhance adipogenic differentiation potential with these inhibitors under normal growth media. As mentioned earlier, bm-MPCs from type 2 diabetic patients show high expression levels of *Cebpa*, which intrinsically favours adipogenic differentiation[61]. Although, ALK5 inhibition did not change the

*PPARG2* levels, inhibiting JNK significantly induced *PPARG2* in the cells (Fig. 6c). A dampened response was also seen with TAK1 inhibitor. These results suggest that TGFB restricts adipogenic differentiation in bm-MPCs by maintaining non-canonical TAK1 activity.

**TGFB1 induces canonical Wnt signaling in bone marrow-derived mesenchymal progenitor cells**. We performed a microarray analysis to understand the mechanisms of TGFB1-mediated inhibition of adipogenesis and to identify targetable pathways. We cultured bm-MPCs in normal growth media (Control), adipogenic differentiation media (ADP), adipogenic differentiation media with TGFB1 (TGFB1), and adipogenic differentiation media with TGFB1 + TAK1 inhibitor (TAK1i). With a threshold of $p = 0.05$ (Fig. 7a), we found a total of 193 genes that were significantly higher in ADP, suppressed in TGFB1, and rescued with TAK1i. On the other hand, 154 genes were downregulated in ADP, increased in TGFB1, and normalized by TAK1i. From these 347 identified genes, Control and TGFB1 groups, and ADP and TAK1i groups displayed similar gene expression signatures and were clustered together (Fig. 7b).

To identify biological process targets, we performed a GO analysis of the identified genes. We found that the upregulated genes in ADP (and rescued TAK1i in TGFB1-treated cells) are largely involved in fatty acid and lipid metabolism and regulation, with 38 GO biological processes identified with a $p < 0.05$. This is not unexpected in stem cell differentiation studies where most of the genes and pathways would be related to the phenotype of cells being derived. More interestingly, the downregulated genes are involved in various biological processes, including osteoblast differentiation and the Wnt signaling pathway, with 19 GO biological processes identified with a $p < 0.05$ (Supplementary Table S2, S3).

To further extract biological insight from the microarray dataset, we performed a Gene Set Enrichment Analysis (GSEA) for hallmark gene sets (v7.4; 50 gene sets) comparing two groups at a time. The groups that were compared were (1) Control vs ADP (Fig. 7c, Supplementary Table S4), (2) ADP vs TGFB1 (Fig. 7d, Supplementary Table S5), and (3) TGFB1 vs TAK1i (Fig. 7e, Supplementary Table S6). Control and TGFB1 groups were indicated with negative normalized enrichment scores, whereas ADP and TAK1i groups were indicated with positive normalized enrichment scores. Next, common enriched gene sets were identified within the groups that displayed similar gene expression signatures. A total of 12 gene sets were enriched in the control and TGFB1 groups, and 10 gene sets were enriched in the ADP and TAK1i groups (Figs. 7c–e). Of the 12 gene sets that were identified in the control and TGFB1 groups, the *Wnt Beta Catenin Signaling* was of great interest, as our laboratory has previously identified that high-glucose induced adipogenic differentiation of bm-MPCs involves the switch from canonical to noncanonical Wnt signaling[6].

To investigate the potential Wnt signaling mechanism, downstream of the TGFB pathway, we exposed bm-MPCs to TGFB1 in adipogenic media and examined mediators involved in the canonical Wnt pathway. Apart from *TCF7L1*, other canonical Wnt mediators, including *CTNNB1* and *LEF1*, were significantly induced with TGFB1 exposure (Supplementary Fig. S21a). Moreover, TLE1 protein, a co-repressor in the Wnt pathway, localization in nuclei of the cells was lost with TGFB1 exposure (Supplementary Fig. S21b), supporting activation of the canonical Wnt pathway. Exposure to TGFB1 in normal growth media was able to significantly increase *CTNNB1* while decreasing *CCND1* mRNA (Supplementary Fig. S21c). Furthermore, TAK1 inhibition in the presence of TGFB1 and adipogenic differentiation

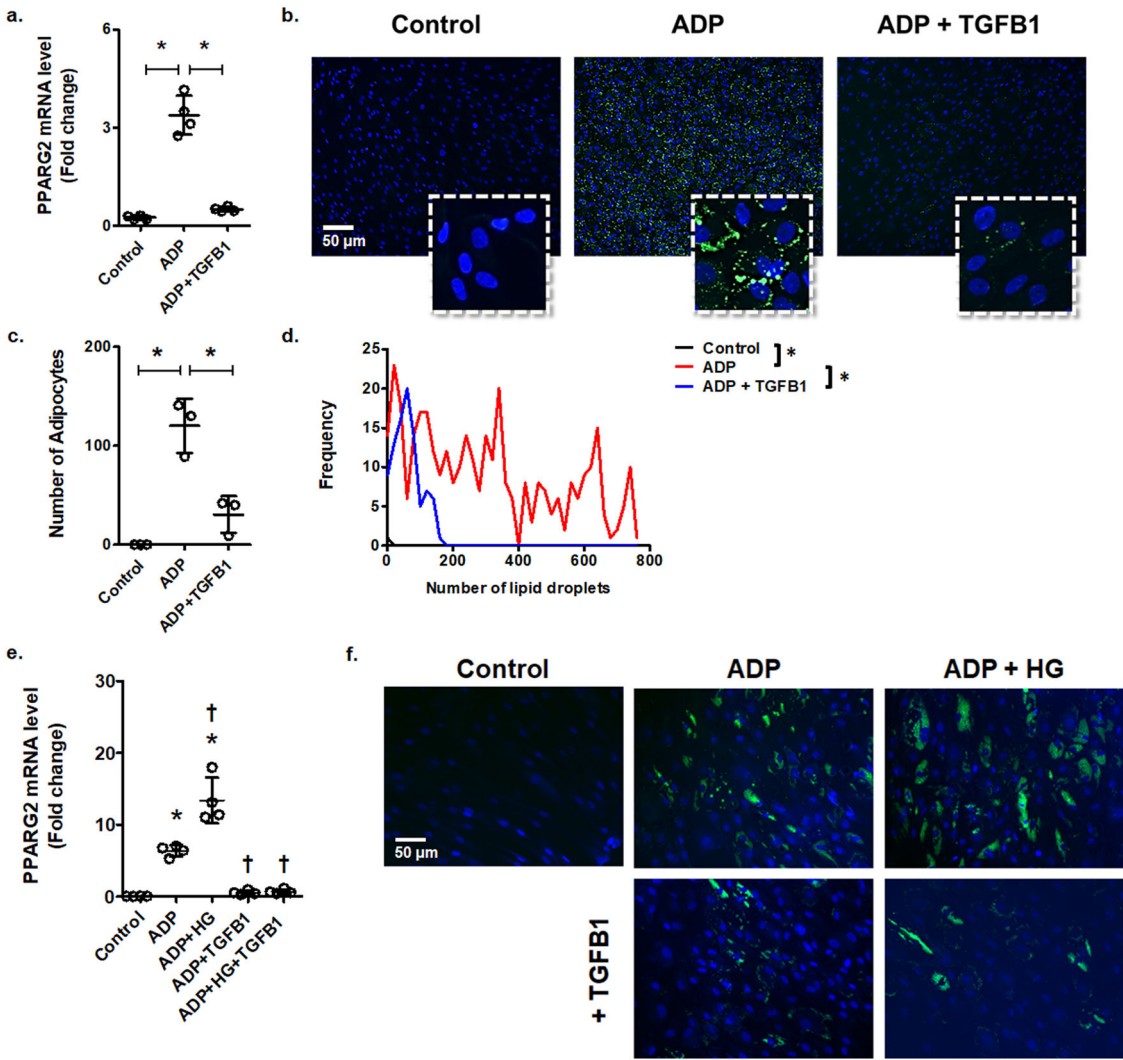

**Fig. 5 TGFB1 exposure inhibits adipogenic differentiation of bone marrow mesenchymal progenitor cells.** Human bone marrow-derived progenitor cells (bm-MPCs) were induced to differentiate in an adipogenesis-inducing media (ADP), with or without TGFB1 (10 ng/mL) for 72 h. **a** mRNA levels of *PPARG2* in bm-MPCs [Data normalized to *ACTB, GAPDH,* and *RPLP0;* Mean ± SD; $n = 4$; each data point represents an independent sample; ANOVA followed by Bonferroni post hoc analysis: *$p < 0.05$]. **b** bm-MPCs, treated as indicated in panel **a**, were stained with LipidTOX (green) to detect intracellular lipid accumulation. Cells were counterstained with DAPI (blue) [scale bar = 50 μm]. Inserts showing higher magnification. The number of adipocytes and frequency of lipid droplets were measured by *CellProfiler* [Mean ± SD; $n = 3$ for panel **c**, 3 images per replicate were measured for panel **d**; ANOVA followed by Bonferroni post hoc analysis: *$p < 0.05$]. **e** bm-MPCs were cultured in control, adipogenic media (ADP), or ADP supplemented with TGFB1 (10 ng/mL) or high glucose (HG; 25 mmol/L) for 7 days. mRNA levels of *PPARG2* were measured [Data normalized to *ACTB;* Mean ± SD; $n = 4$; each data point represents an independent sample; ANOVA followed by Bonferroni post hoc analysis: *$p < 0.05$ compared with control, †$p < 0.05$ compared with ADP]. **f** Detection of intracellular lipid accumulation in bm-MPCs by LipidTOX (green) staining. Cells were counterstained with DAPI (blue) [scale bar = 50 μm]. Cells were treated as indicated in Panel **e**.

signals significantly increased *PPARG2* while significantly decreasing *WISP1* (canonical Wnt target gene) levels (Supplementary Fig. S21d, e). This suggests that TGFB1 may activate the canonical Wnt signaling pathway in bm-MPCs to regulate cell differentiation.

As TGFB1 inhibits adipogenic differentiation, it is possible that one underlying mechanism involves skewing the differentiation of bm-MPCs towards the osteogenic lineage. In addition, the osteoblast pathway was identified in the microarray analysis in cells exposed to TGFB1. Therefore, we examined the effect of exogenous TGFB1 exposure on osteogenic differentiation of bm-MPCs. Previous studies have shown that TGFB1 can regulate mesenchymal cell differentiation[66]. No changes were seen in the osteogenesis-associated *RUNX2* transcription factor (Supplementary Fig. S22a). However, *SP7* was induced by TGFB1. In

addition, the late marker of osteoblast differentiation, *BGLAP* (also known as Osteocalcin) was significantly downregulated. Staining of cells with Alizarin Red S, which marks mineralization, showed reduced levels in cells exposed to TGFB1 (Supplementary Fig. S22b, c). Hence, these results suggest that TGFB1 may prime bm-MPCs to differentiate into osteoblasts, but cells do not reach maturation, possibly due to continued exposure to TGFB1.

## Discussion
Our studies have discovered that diabetes mediates early structural alterations in the bone marrow. The key findings of our study include: (1) 1 month of diabetes is sufficient to observe enhanced tibia marrow adipogenesis in mice, (2) femurs may be more resistant to diabetes-mediated adipogenesis compared with

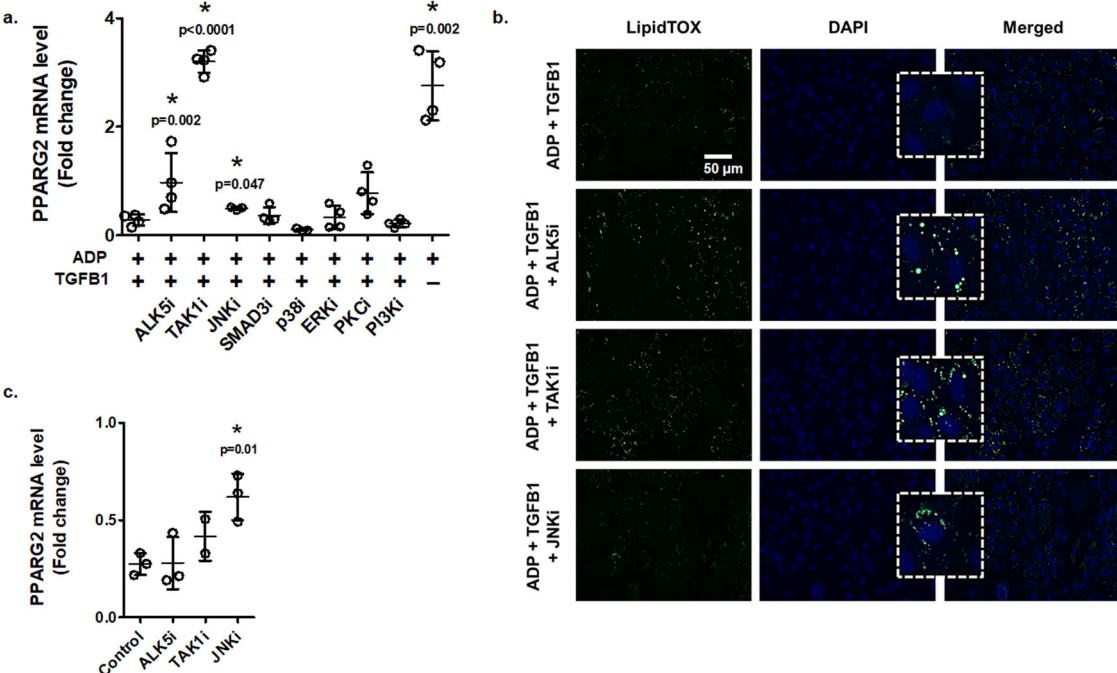

**Fig. 6 TAK1-JNK axis mediates TGFB1 signaling to inhibit mesenchymal progenitor cell differentiation into adipocytes.** Bone marrow-derived mesenchymal progenitor cells (bm-MPCs) were cultured in adipogenic media (ADP) with TGFB1 (10 ng/mL), and various inhibitors of the TGFB signaling pathway. All inhibitors were tested at 10 μmol/L concentration and for 72-hour exposure. **a** Levels of *PPARG2* mRNA in bm-MPCs [Data normalized to *ACTB*, *GAPDH*, and *RPLP0*; Mean ± SD; *n* = 4 for all treatments except ADP + TGFB1 + JNKi and ADP + TGFB1 + p38i which were 3; each data point represents an independent sample; two-tailed student's *t*-test: *$p < 0.05$ compared with adipogenic media and TGFB1 (ADP + TGFB1)]. **b** bm-MPCs, treated as indicated in panel **a**, were stained for intracellular lipid accumulation by LipidTOX (green). Cells were counterstained with DAPI (blue) [scale bar = 50 μm]. Inserts showing higher magnification. **c** bm-MPCs cultured in control (DMEM) media supplemented with TAK1 inhibitor (TAK1i; 10 μmol/L) or JNK inhibitor (JNKi; 10 μmol/L) for 72 h were analyzed for *PPARG2* mRNA levels [Data normalized to *ACTB*, *GAPDH*, and *RPLP0*; Mean ± SD; *n* = 3 control, 3 ALK5i, 3 JNKi, and 2 TAK1i; each data point represents an independent sample; two-tailed student's *t*-test: *$p < 0.05$ compared with control].

tibiae, while both are susceptible to reduced osteogenesis, and (3) TGFB signaling pathway is suppressed in bone tissues of diabetic mice. We further show that enhanced marrow adipogenesis precedes the depletion of marrow-resident stem cells. The main findings of the study are summarized in Fig. 8.

STZ-induced hyperglycemia in rodents generates readouts that mimic human diabetic complications, including skeletal fragility and enhanced adiposity in the bone marrow[67]. Moreover, microangiopathy, neuropathy, and stem cell rarefaction are observed in the bone marrow of diabetic patients and experimental models[6,8,19]. Our studies show increased adipocytes in the tibia to be associated with suppressed TGFB pathway. A recent study investigated the effect of TGFB signaling pathway in bone tissues during development by deleting *Tgfbr2* using a doxycycline-repressible *Sp7* (osterix)-Cre transgene (*Osx-Cre*) in mice 3 weeks of age. These mice show a significant reduction in growth, body weight, bone mass and quality[52]. Moreover, marrow adiposity was significantly enhanced in *Osx-Cre, Tgfbr2fl/fl* mice, where osmium staining and Oil Red O-positive cells in the bone marrow increased almost 80-fold. Therefore, *Tgfbr2* deletion in the bone marrow mirrors diabetes-induced changes such that reduced expression of the TGFB pathway, including the downstream mediators, may remove the suppression and allow bm-MPCs to readily differentiate into adipocytes.

Although our studies focused on the type 1 model of diabetes, extensive investigations by researchers show expansive bone marrow adiposity in type 2 diabetic patients[19,61], and experimental models of type 2 diabetes[61]. It is unknown whether suppressed *Tgfb1* mediates enhanced adipogenesis in the bone marrow in type 2 diabetes. No changes in *Tgfb1* were seen in

adipocytes isolated from the marrow of type 2 diabetic patients compared to cells from non-diabetic patients[61]. It is quite possible that hyperinsulinemia and adipokine-mediated processes may be involved in type 2 diabetes. In support of this notion, bm-MPCs from type 2 diabetic patients express high levels of adipogenesis-associated *Cebpa*[61]. Furthermore, bone marrow adipocytes from type 2 diabetic patients, but not healthy subjects, produce paracrine factors that enhance adipogenesis in MPCs[61]. Whether this represents an enhanced self-propagating[68,69] property of bone marrow adipocytes from diabetic patients or a TGFB1-independent mechanism is an exciting avenue to explore in future studies.

Another interesting finding of our study is the reduced expression of *Cxcl12* in the marrow of diabetic mice. A recent study has shown that purified bone marrow adipocytes in mice express high levels of *Cxcl12*[70]. It is possible that the reduced *Cxcl12* expression seen in marrow flush samples in our study is partly due to the loss of adipocytes, which express *Cxcl12*, during centrifugation and lysis. However, when examined with other experimental evidence presented, there does appear to be an overall reduction in marrow *Cxcl12* expression. For example, femur tissues contained almost no adipocytes in the control and STZ-induced diabetic mice at the 1-month time point but showed a reduced level of *Cxcl12* in the diabetic mice. Furthermore, staining of intact bone tissues also showed reduced CXCL12 levels in tissues harvested from diabetic mice. It is possible that CXCL12 suppression is related to disrupted TGFB signaling in diabetes. In support of this potential link to TGFB, exposure of hepatocytes to recombinant TGFB1 has been shown to upregulate *CXCL12*[71]. However, empirical evidence is needed in bone tissues

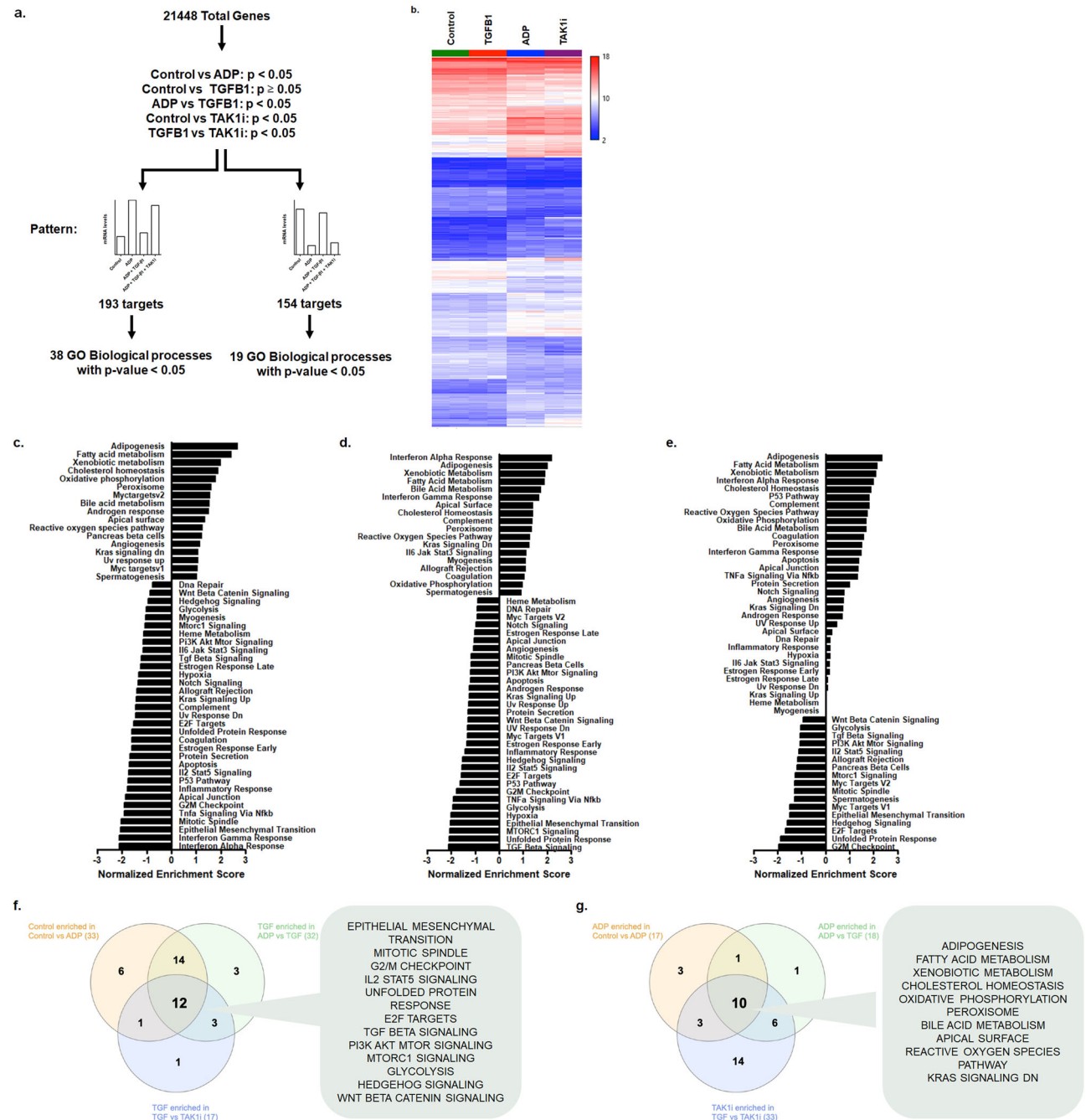

**Fig. 7 Identifying TGFB-responsive genes in bone marrow-derived mesenchymal progenitor cells. a** A threshold of $p = 0.05$ was used to identify differentially regulated genes between the groups. The patterns sought, either (1) genes upregulated with adipogenic media with/without TAK1 inhibitor (TAK1i), which is normalized with TGFB1; or (2) genes downregulated with adipogenic media with/without TAK1i, which is normalized with TGFB1. The identified targets were further analyzed with the DAVID informatics tool to identify significant biological processes. **b** A heatmap of the differentially expressed genes (347 genes) from the Clariom S Assay ($n = 2$ for each group) that clusters Control and TGFB1; and ADP and TAK1i. The gene set enrichment analysis for groups: **c** Control versus adipogenic differentiation; **d** adipogenic differentiation versus adipogenic differentiation with TGFB1; and **e** adipogenic differentiation with TGFB1 versus adipogenic differentiation with TGFB1 and TAK1 inhibitor. A positive normalized enrichment score indicates gene sets that are enriched in bm-MPCs exposed to adipogenic differentiation media (**c, d**) and adipogenic differentiation media with TGFB1 and TAK1 inhibitor (**e**). A Venn diagram illustrating the enriched pathways found in control and TGFB1 (**f**) and adipogenic differentiation and TAK1 inhibitor (**g**). The list of pathways is the common pathways found in all three comparisons.

and bone marrow-derived cells. Nonetheless, reduced CXCL12 level in the marrow is of great significance based on our current knowledge of the potential role of CXCL12[56]. In vitro, CXCL12 acts as a potent chemoattractant for primitive bone marrow CD34+CD38− cells that include hematopoietic stem cells[72–74]. In vivo, CXCL12 has been shown to promote the engraftment of

transplanted hematopoietic cells in the bone marrow and subsequent hematopoietic reconstitution[75]. In diabetic mice, granulocyte-colony stimulating factor fails to reduce *Cxcl12* in the bone marrow, while the levels were efficiently downregulated in non-diabetic mice[76]. However, it should be noted that control mice had significantly higher starting levels of *Cxcl12* compared

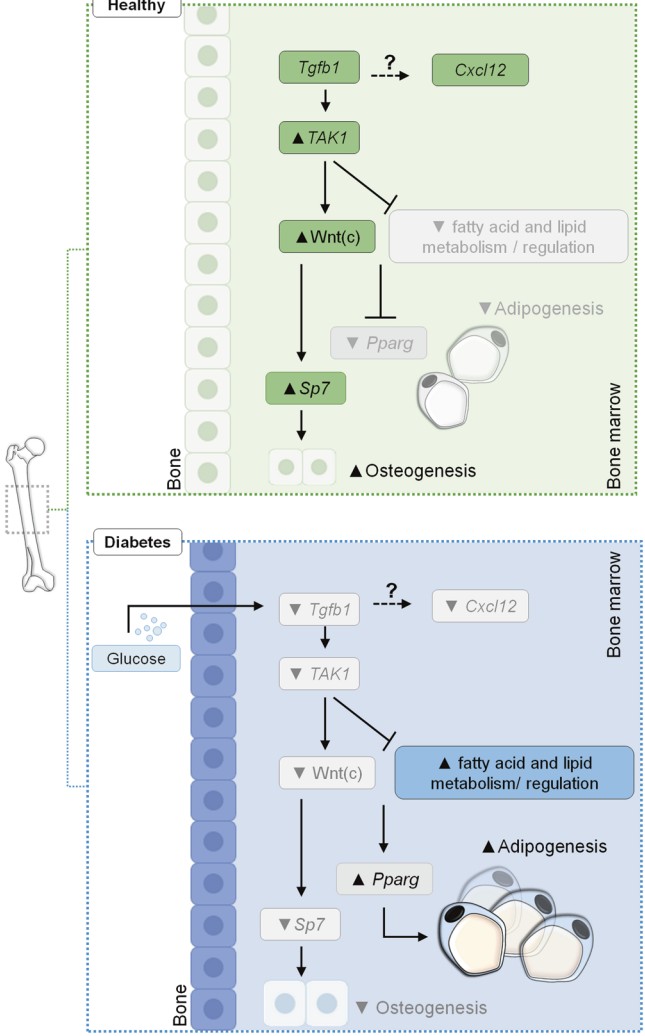

**Fig. 8 Schematic illustrating the working model of diabetes-induced bone marrow dysfunction.** Transforming growth factor-beta (TGFB) signaling in healthy bone tissues maintains a balance between osteoblastogenesis and adipogenesis, in collaboration through canonical Wnt signaling. At least for adipogenesis, TGFB restricts differentiation of marrow progenitor cells through the non-canonical TGFB-activated kinase 1 (TAK1) mechanism. Elevated glucose levels in diabetes suppress the TGFB signaling pathway, which, through alleviation of TAK1, leads to *Pparg* induction and expression of fatty acid- and lipid metabolism-regulating genes to favour adipogenesis. Dashed black arrows indicate linkages that require further investigation.

with diabetic mice. Therefore, reduced basal expression of CXCL12 may potentially participate in hematopoietic stem cell mobilization seen in diabetic patients.

Our studies temporally place TGFB disruption before Wnt alteration in diabetic mice and cultured bm-MPCs. We show that the TGFB1 pathway regulates canonical Wnt signaling in bm-MPCs. Furthermore, TGFB1 exposure enriched the *Wnt Beta Catenin Signaling* gene sets in our profiling. In pre-adipocytes, Wnt signaling decreases as cells differentiate into mature adipocytes, and the expression of CTNNB1 shows an inverse relationship with levels of CEBPA, a key regulator of adipogenesis[77]. In our study, the TGFB1-mediated activation of the canonical Wnt pathway was evident with the loss of nuclear expression of TLE1 along with the induction of *CTNNB1*.

In summary, our studies identified bone marrow alterations as one of the early complications of diabetes. We show enhanced

bone marrow adiposity in tibiae of mice to be associated with suppressed TGFB signaling, a known regulator of cell differentiation. We explored the functional significance of disrupted TGFB signaling using bm-MPCs and showed that TGFB1-mediated TAK1-JNK axis activation plays an important role in preventing adipogenic differentiation. Identifying avenues to activate the noncanonical TGFB signaling in the marrow may be important to preserving bone integrity in diabetes and the marrow regenerative stem cells.

## Methods

**Diabetic mouse modelling**. All mouse studies were initiated after receiving approval from the Animal Care and Veterinary Services at Western University (Animal Use Protocol: #2019-125).

Four-week-old male C57BL/6 mice (Charles River Canada) were obtained and allowed to acclimate for 1 week. Mice were kept on a light/dark (12 h/12 h) cycle at 23 °C and received standard rodent chow and water *ad libitum*. Mice were divided into two groups, diabetic and non-diabetic, by ensuring the body weights were similar between the two groups. Diabetes was induced by multiple low-dose intraperitoneal injections of freshly prepared streptozotocin (STZ; 50 mg/kg in 0.1 M citrate buffer, pH 4.5; Sigma-Aldrich; Cat# S0130) using 25 G needles, for 5 consecutive days[78]. Non-diabetic control mice received an equivalent volume of freshly prepared citrate buffer alone. Body weights were measured daily for the duration of the study. Mice were also monitored daily for body condition scores and overall health. Hyperglycemia was confirmed 1 week after the last STZ injection by measuring non-fasting blood glucose levels from the tail vein, using a glucometer (CONTOUR Next One, Ascensia Diabetes Care). This day was considered day zero (d0). Mice with non-fasting blood glucose levels greater than 11.1 mmol/L (>200 mg/dL) were considered hyperglycemic/diabetic. Mice that did not meet the threshold were measured again after 2 days to confirm hyperglycemia. All mice injected with STZ exhibited hyperglycemia and were included in the study. Mice in non-diabetic and diabetic groups remained at a body condition score of 3 (well-conditioned). Each group contained seven mice.

One day prior to euthanasia, blood glucose levels were measured again. Following $CO_2$ euthanasia, mice were monitored for cessation of breathing/heartbeat. An incision in the chest wall was made immediately to reach the heart. A small slit was made into the right atrium. Mice were then perfused with 5 mL of cold phosphate-buffered saline (PBS; Fisher Scientific; Cat# BP399-1) by inserting a 28 G needle into the left ventricle. Femurs and tibiae were then removed and freed from soft tissue. Limbs from one side were used for histology, and the other side was used for preparing flush samples. For histological analyses, the bones were fixed in 10% neutral buffered formalin (Sigma-Aldrich; Cat# HT501128) overnight at room temperature, decalcified in 0.5 mol/L ethylenediaminetetraacetic acid (EDTA; Fisher Scientific; Cat# S311) for 7 days at 4 °C, and rehydrated in 30% sucrose solution (Sigma-Aldrich; Cat# 84100) overnight at 4 °C. Samples were moved to 70% ethanol and processed and embedded in paraffin. All bone tissues were sectioned at a 4-μm thickness to ensure tissues remain affixed to the glass slides (Fisher Scientific; Cat# 1255017). For gene expression analyses, the epiphyses of the femurs and tibiae were removed, and tissues were placed in a 0.5 mL centrifuge tube nested in a 1.5 mL centrifuge tube. A small hole was created at the bottom of the 0.5 mL tubes with an 18G needle, prior to adding the tissues. Tubes containing bone tissues were then centrifuged at 10,000 x*g* for 15 s to flush the bone marrow. This method of preparing cell suspensions removed all cellular content from the marrows, as can be detected by histological analysis before and after the procedure. The resulting flush was stored in a cell freezing media solution (90% fetal bovine serum and 10% dimethyl sulfoxide; Millipore; Cat# S-002-5F) at −20 °C until further analysis. For RNA isolation from the marrow cells, the flush samples were quickly thawed to 37 °C in a water bath and centrifuged at 300 xg to pellet the cells. Cells were then suspended in RLT cell lysis buffer (RNeasy Plus Mini Kit; Qiagen, Cat# 74134).

In addition to the bone tissues, other tissues from mice were collected. Specifically, eyes, heart, lung, kidney, liver, pancreas, and epididymal adipose tissues were removed for histology. The tissues were fixed in 10% neutral buffered formalin (Sigma-Aldrich; Cat# HT501128) overnight at room temperature prior to paraffin embedding. Non-bone tissues were sectioned at 5-μm thickness for immunostaining and other histological studies.

A longer-term diabetes study was also performed. Following the same procedure as indicated above, non-diabetic and STZ-induced diabetic mice were followed up at 2 months. Femurs and tibiae were removed and processed as indicated above. Bone marrow flush samples were also prepared for gene expression studies. Select target organs of diabetic complications such as the heart, kidney, and eyes were also harvested for histological studies.

**RNA isolation, mRNA profiling & Quantitative PCR**. Total RNA from cultured cells and marrow flush samples was isolated using RNeasy Mini Plus Kit (Qiagen; Cat# 74134). RNA from formalin-fixed paraffin-embedded (FFPE) epididymal adipose tissue sections (total of 20-μm thickness) was collected using the PureLink FFPE RNA Isolation Kit (Thermo Fisher; Cat# K156002). Total RNA was

measured using Qubit RNA Broad Range Assay (Thermo Fisher; Cat# Q10210) or the Qubit RNA High Sensitivity Assay (Thermo Fisher; Cat# Q32852) in a Qubit Fluorometer (Thermo Fisher; Cat# Q32857). cDNA was synthesized using iScript cDNA Synthesis Kit (Bio-Rad Laboratories; Cat# 1708841). qPCR reactions for SYBR-based amplification and detection consisted of 10 μL RT² SYBR Green qPCR Mastermix (Qiagen; Cat# 330502), 2 μL of both forward and reverse primers (Supplementary Table S7 and S8) at 10 μmol/L concentration, 2 μL cDNA, and 6 μL nuclease-free $H_2O$ (Ambion; Cat# AM9937). For TaqMan-based chemistry, the reactions consisted of 5 μL TaqMan Fast Advanced Master Mix (Thermo Fisher; Cat# 4444963), 0.5 μL of the gene probe (Supplementary Table S7 and S8), 1 μL cDNA, and 3.5 μL nuclease-free $H_2O$. All reactions were run on CFX Connect Real-Time PCR Detection System (Bio-Rad Laboratories). Target gene mRNA data were normalized to housekeeping genes identified in figure legends. For most studies, the geometric mean of multiple housekeeping genes was used for normalization. SYBR Green-based reactions were performed for 40 cycles using the following temperature profiles: 95 °C for 15 s (initial denaturation), and 60 °C for 60 s (annealing and extension). TaqMan-based reactions were performed for 40 cycles using the following temperature profiles: 95 °C for 3 s (initial denaturation), and 60 °C for 30 s (annealing and extension). Data were analyzed by *CFX Manager* software (Bio-Rad; Cat# 1845000) using the normalized (ΔΔCT) method.

**Histomorphometric analyses**. Histomorphometric analyses were performed on mouse retina, epididymal fat, and bone tissues. The thickness of retinal layers and bone lengths were quantified using *QuPath*[79]. The frequency of the area of adipocytes found in the white epididymal adipose tissue was analyzed using *Adiposoft*, an automated software[80]. The number of nuclei per area in tissues was determined by *QuPath*[79]. Bone marrow analysis was conducted while considering the guidelines and standards previously reported[81]. The bone marrow histological sections were analyzed using *MarrowQuant*[36] and *QuPath*[79].

**Hematoxylin & Eosin (H&E), Picro-Sirius Red, and PAS staining of mouse tissues**. Deparaffinized tissue sections were stained using hematoxylin (Leica Biosystems; Cat# 3801561) and eosin (Leica Biosystems; Cat# 3801601) for routine histological analysis. Picro-Sirius Red (Abcam; Cat# ab246832) was used to mark collagen deposition. Periodic Acid-Schiff (PAS) stain kit (Abcam; Cat# ab150680) was used to detect the presence of carbohydrates and carbohydrate compounds such as polysaccharides, mucin, and glycogen. Liver sections were also subjected to α-amylase (Sigma-Aldrich; Cat# A3176) treatment prior to PAS (PAS-diastase Staining) to remove glycogen. The stained slides were digitalized using Aperio slide scanning system (Aperio Technologies, Inc., Vista, CA) with a 20X and 40X objective and saved in Tagged Image File Format (TIFF). Image analyses were performed with the *ImageJ* software package (National Institutes of Health, Bethesda, MD, USA). Picro-Sirius red-stained area was automatically assessed by means of a macro (https://github.com/northcottj/picrosirius-red). The Picro-Sirius Red-positive area of the specimens was quantified using *ImageJ* software and shown as a percentage of the total section area.

**Immunostaining of tissues**. Tissue sections (all except bone tissues) were deparaffinized in xylenes, hydrated in an ethanol gradient, and subjected to a heat-induced antigen retrieval using Citrate/EDTA buffer (10 mmol/L sodium citrate, 1 mmol/L EDTA, 0.05% Tween-20, pH 6.0) or Tris/EDTA buffer (10 mmol/L Trizma base, 1 mmol/L EDTA, 0.05% Tween-20, pH 9.0) in 2100 Retriever (Electron Microscopy Science, Hatfield, PA). Sections were incubated with rabbit anti-mouse perilipin-1 (PLIN1; Abcam; Cat# ab3526; 1:200 dilution) or insulin (INS; Thermo Fisher; Cat# 15848-1-AP; 1:1000 dilution) at room temperature for 1 h. FITC- or peroxidase-conjugated secondary antibodies (Vector Laboratories, Burlington, ON) were used for detection. Where peroxidase-conjugated secondary antibody was used, AEC (3-amino-9-ethylcarbazole) substrate (Vector Laboratories; Cat# SK-4200) was utilized for detection, followed by hematoxylin counterstaining. For immunofluorescence staining, sections were exposed to Vector TrueVIEW Autofluorescence Quenching Kit (Vector Laboratories; Cat# SP-8400-15) for 5 min, followed by counterstaining with DAPI (Sigma Aldrich; Cat# F6057). Fluorescence images were captured with an Olympus BX-51 microscope equipped with Infinity 3-1 Color CCD camera (Teledyne Lumenera, Ottawa, Canada) and *Infinity Analyze* software (Teledyne Lumenera). AEC-stained slides were digitalized using an Aperio slide scanning system (Aperio Technologies) with a 20X objective and saved in TIFF.

**Bone tissue immunostaining**. Bone tissue sections were deparaffinized in xylenes and rehydrated through an ethanol gradient. Tissue sections were then subjected to a heat-induced antigen-retrieval process using a microwave. Briefly, jars containing the antigen retrieval buffer (10 mmol/L sodium citrate, 1 mmol/L EDTA, 0.05% Tween-20, pH 6.0 or 10 mmol/L Tris, 1 mmol/L EDTA, 0.05% Tween-20, pH 9.0) were placed in the microwave and the buffer was boiled. Tissue sections were then immersed in the jar when the buffer had cooled and was not boiling. This process was repeated until tissues had been subjected to a total of 15 min in a heated buffer. Sections were then blocked in 1% bovine serum albumin (Sigma-Aldrich; Cat# A7906) solution in PBS containing 0.1% Tween-20 (Sigma-Aldrich; Cat# P9416) for 1 h, and then incubated with primary antibody (Supplementary Table S9) for

1 h at room temperature. Fluorophore-conjugated secondary antibodies were applied for 1 h at room temperature. Secondary antibodies included Alexa Fluor 488-conjugated anti-goat (Thermo Fisher; Cat# A-11079), FITC-conjugated anti-rabbit (Thermo Fisher; Cat# F-2765), or FITC-conjugated anti-mouse (Vector Laboratories; Cat# FL-2000). All sections were then exposed to Vector TrueVIEW Autofluorescence Quenching Kit (Vector Laboratories; Cat# SP-8400-15) for 5 min, followed by counterstaining with DAPI (Sigma Aldrich; Cat# F6057). Negative control sections were included for all staining studies and consisted of the same tissue and procedure but with the omission of the primary antibody. Fluorescence images were captured using an Olympus BX-51 microscope. Where fluorescence intensity was measured, sections from all experimental conditions were subjected to the same exposure time when capturing images. Fluorescence staining intensity per area of the sections was performed using *QuPath* and *ImageJ*.

**Bone tissue in situ hybridization**. *Tgfb1* mRNA was detected in mouse bone tissues using RNAscope in situ hybridization probes and RNAscope 2.5 Red Reagent Kit (Advanced Cell Diagnostics; Cat # 322350). Paraffin-embedded tissue slides were baked at 60 °C for 1 h in Lab-Line TempCon Oven (American Scientific Products, Livonia, MI). Slides were deparaffinized in xylenes for 10 min and then incubated in absolute ethanol for 2 min. Endogenous peroxidase activity was blocked by incubating slides in RNAscope Hydrogen Peroxide (provided with Red Reagent Kit) for 10 min. Slides were washed in dH2O and immersed in Custom Pretreatment reagent for bone samples (Advanced Cell Diagnostics; Cat # 300040) at 40 °C for 30 min in HybEZ Oven (Advanced Cell Diagnostics; Cat # 310010). Probes were then added, and slides were incubated at 40 °C for 2 h in HybEZ Oven. Probes included Mm-*Tgfb1* (Cat# 407751) and Bacillus subtilis dihydrodipicolinate reductase (*Dapb*) negative control (Cat # 10043). Signal amplifiers were added in sequence, as recommended by Advanced Cell Diagnostics. Detection was performed with the RED reagent (Advanced Cell Diagnostics; Cat # 322350). Slides were counterstained with hematoxylin and mounted. Brightfield images were taken with an Olympus BX51 microscope. The same slides were imaged using mercury arc illumination and TRITC filter. RNAscope 2.5 Red Reagent is naturally fluorescent around 570 nm (TRITC/CY3).

**Isolation, culture, and differentiation of MPCs**. Human bone marrow-derived mesenchymal progenitor cells (bm-MPCs) were prepared from freshly isolated human bone marrow mononuclear cells (Lonza Inc., Walkersville, MD). Multiple donors were used in this study (Lot# 081032A, 43-year-old Caucasian female; Lot# 081109 A, 43-year-old Hispanic male; Lot# 080511A, 20-year-old Black female; Lot# 080500B, 19-year-old Black male; Lot# 081362 A, 27-year-old Caucasian female; Lot# 080455B, 19-year-old Black male; Lot# 081393A, 19-year-old Black female). Bone marrow mononuclear cells were seeded on tissue culture plates in low glucose Dulbecco's Modified Eagle Medium (DMEM; Thermo Fisher; Cat# 11885084), supplemented with 10% fetal bovine serum (FBS; Lonza, Cat# 14-507 or Thermo Fisher, Cat# 12484028) and 1X PSF (antibiotic-antimycotic solution; Corning, Cat# 30-004-CI).

All in vitro studies were performed with at least four different donor preparations (N) conducted at different times, and more than two experimental replicates (n) each time. All bm-MPC preparations were used before sub-passage 7. To induce adipogenic differentiation, bm-MPCs were plated at a density of 40,000 cells/cm². The next day, cells were exposed to adipogenesis-inducing media for various time points, ranging from 48 h to 7 days (Supplementary Table S10). Adipogenesis-inducing media comprised of DMEM low glucose media supplemented with 1X StemPro Adipogenesis Supplement from StemPro Adipogenesis Differentiation Kit (Thermo Fisher; Cat# A1007001). We did not use the base media provided in the Adipogenesis Differentiation kit as it contained high levels of glucose. Seeding cells at a high density prior to the adipogenesis media exposure was performed to ensure we capture the differentiation phase only and not the early mitotic burst-related changes. Although the formulation of commercial adipogenesis media is proprietary, adipogenesis in human mesenchymal precursors is easily induced in media containing 10% FBS, 10 μg/mL insulin, 1 μM dexamethasone, 0.5 mM isobutylmethylxanthine, and 60 μM indomethacin. However, for consistency in experiments conducted with different donor cells and at different times, the pre-formulated commercially available media was used. We did confirm with the vendor that the media formulation does not contain peroxisome proliferator activated receptor gamma (PPARG) agonists, such as pioglitazone, which would confound the results. Control media consisted of DMEM low glucose media supplemented with 10% FBS. Media was changed every other day.

For some studies, we tested adipogenic differentiation of bm-MPCs in the absence of insulin. For these experiments, we produced an insulin-deficient adipogenesis-inducing media (idADP) containing serum, dexamethasone, isobutylmethylxanthine, and indomethacin (Supplementary Table S1). We then supplemented this base idADP with various test agents including high glucose (25 mM), linoleic acid-oleic acid (LIN/OL; a source of fatty acids), or a combination. Cells were cultured in the test media for 7 days with media change every other day. Phase images were captured on day 7 using Olympus BX-51. Cells were then fixed with paraformaldehyde (Electron Microscopy Sciences; Cat# 15710) for 10 min, and then stained with HCS LipidTOX Green neutral lipid stain

(Thermo Fisher; Cat# H34475) for 30 min, as detailed below. Fluorescence images were captured.

To determine the effect of TGFB1 on bm-MPC differentiation, exogenous human TGFB1 (R&D Systems, Cat# 100-B-001) was added to the adipogenesis induction media (DMEM low glucose media supplemented with 1X StemPro Adipogenesis Supplement) at 10 ng/mL concentration. To determine how TGFB1 mediates its effect in bm-MPC differentiation, downstream signaling proteins that are responsive to TGFB1 were inhibited using commercially available inhibitors (Supplementary Table S11).

To induce osteoblast differentiation in bm-MPCs, cells were plated at a density of 40,000 cells/cm$^2$. After 24 h, cells were exposed to osteogenesis-inducing media for various time points, ranging from 9 days to 21 days (Supplementary Table S10). To prepare osteogenesis-inducing media, DMEM low glucose was supplemented with 1X StemPro Osteogenesis Supplement provided in the StemPro Osteogenesis Differentiation Kit (Thermo Fisher; Cat# A1007201). At the end of the experiment, cells were fixed and stained with Alizarin Red S (Sigma-Aldrich; Cat# A5533), as detailed below.

**Transcriptome-wide gene-level expression profiling of bm-MPCs.** Total RNA from bm-MPCs was isolated using RNeasy Mini Plus Kit (Qiagen; Cat#74134) as detailed below. RNA samples were sent to the Genetic and Molecular Epidemiology Laboratory, David Braley Research Institute (Hamilton, Canada) for human Clariom S Assay (Thermo Fisher; Cat# 902926) with standard input ("Plus" assay). Raw data files were obtained to be viewed using Transcriptome Analysis Console Software (Thermo Fisher).

Microarray data discussed in this publication is deposited in NCBI's Gene Expression Omnibus[82] and are accessible through GEO Series accession number GSE184612[83].

Gene ontology (GO) analysis was performed using the *Database for Annotation, Visualization and Integrated Discovery (DAVID)* online bioinformatics program (https://david.abcc.ncifcrf.gov[84,85]). Gene set enrichment analysis (GSEA) was performed with the *GSEA* software (http://www.broadinstitute.org/gsea/[86,87]). Enrichment score (ES) is the degree to which a gene set is over-represented in the expression dataset. Normalized enrichment score (NES) is the enrichment score that has been normalized across analyzed gene sets. Venn diagrams were created with *InteractiVenn* (http://www.interactivenn.net/index.html[88]) and are not plotted to scale.

**Cell staining**

*Immunostaining.* Cells were plated at a density of 40,000 cells/cm$^2$ on 8-chambered Nunc Lab-Tek Chamber Slide System slides (Thermo Fisher; Cat# 177402). Cells were allowed to attach for 24 h. After various treatments, cells were fixed in cold methanol for 10 min on ice. Cells were then permeabilized with 0.25% Triton X-100 (VWR; Cat# VWRVM143) in PBS before incubating with primary antibody against TLE1 (Abcam; ab183742; 1:200 dilution). Following 1 h of primary antibody incubation, FITC-conjugated secondary antibody (Vector Laboratories; Cat# FI-1000) was applied for 1 h at room temperature. Slides were mounted using a medium containing DAPI (Sigma-Aldrich; Cat# F6057). Images were taken using an Olympus BX-51.

*Lipid Accumulation.* To detect intracellular lipid accumulation in cells, HCS LipidTOX Green neutral lipid stain (Thermo Fisher; Cat# H34475) was used. After various treatments, cells were fixed with 3–4% paraformaldehyde (Electron Microscopy Sciences; Cat# 15710) for 10 min. LipidTOX staining solution was then added for 30 min. Cells were counterstained with DAPI. The number of adipocytes and lipid droplets was measured by *CellProfiler* software[89].

*Mineralization stain.* To detect osteoblast differentiation and mineralization, Alizarin Red S staining was performed. Cells were fixed in 10% neutral buffered formalin for 30 min. A 2% Alizarin Red S solution (Sigma-Aldrich; Cat# A5533) was added for 5 min. Slides were then cleared in 50% xylene and 50% acetone. Slides were mounted for imaging using a water-based mounting media. To quantify the staining, a 10% acetic acid solution was added to each well and heated at 85 °C for 10 min. The solution was allowed to cool, and samples were centrifuged at 20,000 x *g* for 15 min. The supernatant was then transferred to a new tube and sodium hydroxide solution was added to neutralize the acid to a pH between 4.1 and 4.5. Absorbance was measured at 405 nm using Thermo Scientific Multiskan FC Microplate Photometer (Thermo Scientific).

**Statistics and Reproducibility.** All data were expressed as Mean ± standard deviation (SD). A single sex and starting age of the animals was studied. All tissues were analyzed from mice, without exclusion. For cell-based studies, all experiments were repeated in at least 4 biological replicates (donors). Small 'n' represents an independent sample, not a technical replicate or an additional reading of the same sample. An a priori analysis was not performed. Sample size for cell culture studies was based on our previous publications[6,65]. Sample size for in vivo studies was based on previous studies showing gene expression changes[90] and target organ cell composition changes[91] in diabetes. Statistical and graphical analyses were performed using GraphPad Prism 7 and Microsoft Excel (basic statistical functions for

descriptive statistics only). Data were tested for normality. When comparing parametric data from two groups, a two-tailed Student's unpaired t-test was used. For multiple comparisons, an analysis of variance (ANOVA) followed by Bonferroni post hoc analysis was performed. P values <0.05 were considered statistically significant.

**Reporting summary.** Further information on research design is available in the Nature Research Reporting Summary linked to this article.

## Data availability

All data figures show raw/individual data points. Source data for the main figures is provided as Supplemental Data 1. Gene profiling data is deposited in public repository and available to researchers (GSE184612). Data for Supplementary Figures are available from the corresponding authors on reasonable request. Reagents and other research material used in this study, if not depleted, will also be provided to other researchers upon request.

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

## Acknowledgements

This work was supported by grants from the Canadian Diabetes Association (ZAK; OG-3-13-4034-ZK), Western Strategic Support (ZK), and Lawson Health Research Institute (ZAK). JJYK was funded, in part, through support from the Ontario Graduate Scholarship, Frederick Winnett Luney Graduate Scholarship, and Western's Interdisciplinary Development Initiative in Stem Cells and Regenerative Medicine Graduate Student Training Award. Authors thank Ms. Linda Jackson-Boeters in the Department of Pathology and Laboratory Medicine at Western University for tissue embedding and sectioning. The authors also thank Dr. Biao Feng in the Department of Pathology and Laboratory Medicine at Western University for initiating diabetes and maintaining mice in the 2-month follow-up study.

## Author contributions

All experiments included in this study were performed by J.J.Y.K. J.J.Y.K. also designed the experiments and analyzed the results. Z.A.K. and C.J.H. were involved in the conceptual design of the study, interpretation of the data, and editorial review. C.J.H. read all staining images for histopathological assessment. All authors were involved in writing of the manuscript and approve the final submission.

## Competing interests

The authors declare no competing interests.

## Inclusion and Ethics

All researchers involved in the study are listed as authors or acknowledged, as appropriate.
