## [Peer Review File · Communications Biology]

Reviewers' comments:

Reviewer #1 (Remarks to the Author):

This paper reports data regarding the adipogenic features of tibial bone marrow in mice made diabetic using streptozotocin. It also reports the downregulation of TGF β signalling in BM and the effect of high glucose on BM stromal cells to commit to adipogenesis, possibly through a mechanism involving the TGF β signalling pathway.

They report that the adipogenic phenotype is limited to the tibia at the early stage of diabetes before structural changes occur, therefore they conclude that this complication precedes other systemic complications.

The study includes a large number of histological and molecular data, well illustrated by the figures. The experiments seem to be properly conducted. I have the following comments that I hope can help the authors in improving their contribution.

1. I agree that there is a dispute about the best diabetic model. However, the authors seem excessively concerned of this aspect to the extent to dedicate a large part of the introduction on the appropriateness of the model instead of identifying the clinical importance of adipogenesis in the diabetic BM, the possible implications and interaction with other adipogenic sites, and the working hypothesis and objectives.

2. In fact, they simply state that the main scope was to examine the effect of STZ-induced diabetes in C57BL/6 mice, with the goal of identifying early changes that may mediate increased bone marrow adipogenesis. They indeed found the adipogenic accumulation is characterized by excess small adipocytes, but the assumption this is antecedent to complications remains questionable. They used an histological approach but they cannot exclude that functional alterations were already present at the same stage. For instance, it is known that type 1 diabetic children have precocious alterations in left ventricular diastolic function. In a paper published in *Diabetes Care*, Suys et al. (*Diabetes Care* 27: 1947–1953, 2004) provided evidence that children and adolescents with type 1 diabetes have altered cardiac structure and function compared with age-matched individuals without diabetes. Subjects included in the study had no cardiac signs or symptoms or diabetes complications and were not taking medications known to modify cardiac structure or function. The most striking findings were in the girls with type 1 diabetes. Similarly, seminal studies showed the exercise proteinuria is present in children with type 1 diabetes (*Diabetologia*. 1981 Nov;21(5):495-7. doi: 10.1007/BF00257792). Therefore, I suggest the authors to perform functional studies on target organ damage and revisit/discuss these functional data with current evidence of early organ damage.

3. In addition, the clinical relevance of this early phenomenon remains obscure. Can the authors elaborate more the concept? Do they think that the transformation drives those other complications (see our review *Bone marrow fat: friend or foe in people with diabetes mellitus?* Santopaolo M, Gu Y, Spinetti G, Madeddu P. *Clin Sci (Lond)*. 2020 Apr 30;134(8):1031-1048. doi: 10.1042/CS20200220); or represents an independent phenomenon reflective of a selective susceptibility of BM stromal cells as compared with other cells in different organs?

4. The study describes a plethora of molecular modifiers, and then focuses on TGF β , which appears to be the second pillar of this paper. Although the data in BM stromal cells seem confirmatory of this implications, I remain concerned about the lack of information regarding the intrinsic mechanism causing the TGF β downregulation. For instance, TGF β could be regulated at transcriptional levels or inactivated through formation of latent complex, that is biologically inactive. How high glucose or lack of insulin interfere with these mechanisms?

5. This data on TGF β is also different from our report of enhanced adipogenesis in type 2 diabetic mice (we could not find any change in mRNA levels of TGF β) that the authors did not include in their reference section, but that we think would be relevant to report and discuss/elaborate vs their data (*MCP-1 Feedback Loop Between Adipocytes and Mesenchymal Stromal Cells Causes Fat Accumulation and Contributes to Hematopoietic Stem Cell Rarefaction in the Bone Marrow of Patients With Diabetes*. Ferland-McCollough D, Maselli D, Spinetti G, Sambataro M, Sullivan N, Blom A, Madeddu P. *Diabetes*.

2018 Jul;67(7):1380-1394. doi: 10.2337/db18-0044. Epub 2018 Apr 27. PMID: 29703845).

5. To support their conclusion on TGF β , inducers of TGF β 1 could be used in vivo.

6. STZ was given according to standard optimized procedure (low repeated doses). Therefore, a possible toxic effect is unlikely responsible for the changes observed in BM. Nonetheless, they found the the animals have a delayed growth. Recent studies have shown that caloric restriction in young, growing mice is associated with impaired skeletal acquisition, low leptin and IGF-1 levels, and high marrow adiposity (J Bone Miner Res. 2010 Sep; 25(9): 2078–2088. Published online 2010 Mar 12. doi: 10.1002/jbmr.82). How can they exclude that the adipogenic change is related to a dietary defect and/or malabsorption of nutrients in the STZ group? To understand if lack of insulin is the driver of the observed BM changes, rescue experiments with insulin could be very important.

7. I also noted that BM of control animals receiving citrate vehicle contains large amount of fat. STZ is prepared in cold citrate buffer at a pH of 4.5 to enhance stability. The authors should consider the possibility that the acidity of the injected citrate had no influence on local pH in BM with synergistic effect on STZ induced damage. In fact, we found that even minimal acidosis in the BM can cause osteoclastogenesis and BM remodelling through the Activation of Transient Receptor Potential Cation Channels (10.1038/srep30639)

8. Regarding statistics, I suggest to confirm that the authors used an adjustment for multiple comparisons.

In conclusion, the study is accurate and rich of data but suffers from an unfocused presentation and weakness in the demonstration of main hypothesis. Is the BM a early driver of diabetes complications? Does TGF β reconstitution correct the phenotype? Is the adipogenesis dependent on insulin?

Reviewer #2 (Remarks to the Author):

The work of Kum and colleagues explores a relevant topic in diabetology, addressing the remodeling of the bone marrow in diabetes.

The authors studied the effects of hyperglycemia on the remodeling of the bone marrow in terms of enhance adipogenesis. These findings are not entirely new, as it has been shown that type 1 diabetes is associated with increased adiposity in the bone marrow (PMID: 32345753). The authors should refer to this work when commenting their findings.

The work has some merit, though. In particular the quality of histological images is excellent. However, some flaws and major issued must be addressed.

Major issues

- The authors shall greatly condense the first paragraph of the "Results". Indeed, it is well established that the streptozotocin-induced diabetic mice do not develop some complications unless after > 2 month of hyperglycemia. Furthermore, other groups have already showed that 1 month of hyperglycemia is sufficient to induce robust alterations in the bone marrow (PMID: 32345753, 23663738, 32316750).

- The authors analyzed epididymal fat and reported an increased number of adipocytes, which is suggestive of hyperplasia. However, this is highly unlikely in the setting of this model of diabetes. Indeed, streptozotocin-induced diabetes induces a massive lipodystrophic phenotype due to the lack of insulin and a profound atrophy of the adipose tissue. This is compatible with the gene expression profile where all the major adipose tissue related genes are downregulated. The increased number of adipocytes is therefore the result of lipolysis due to counterregulatory mechanism activated by the lack of insulin. The authors should revise their conclusion from this set of data.

- Line 208. The authors claim that diabetes does not alter hematopoietic areas, confirming "previous reports". However, diabetes is known to modify hematopoiesis triggering myelopoiesis. Furthermore, diabetes triggers neuropathy and capillary rarefaction (PMID: 24270983, 26358583, 20042708). Thus,

the authors should better explain this sentence.

- Line 2016. While, Sca1 is a stemness marker for HSC, it is also expressed by stromal cells. A flow cytometry approach to quantify the stem cell compartment is more suited for such quantification. Furthermore, diabetes has been associated with some changes in the HSC compartment in previous works. The authors should comment about that.

- In which cells of the bone marrow the TGF- β signaling is reduced in diabetes in vivo? Mesenchymal stromal cells? The authors should show some data about this point in vivo, using immunofluorescence or cell sorting.

- Line 256: The authors should check also for other niche factors like Vcam and Angpt1.

- The authors should consider that adipocyte in the bone marrow have higher expression of Cxcl12 compared to other cells (PMID: 32345753). Therefore, the actual amount of Cxcl12 in the tibiae of diabetic mice might be higher than non-diabetic mice. Indeed, when you flush the bone marrow by centrifugation, and you resuspend the pellet, you'll inevitably loose the adipocytes, unless you collect them separately or lyse the pellet without resuspending it. The authors should consider that when they comment their data.

- Line 436. The authors did not study mobilization so the claim about the role of their data in the mechanism of diabetic stem cell mobilopathy is not supported. Despite the reduced level of Cxcl12 in the bone marrow, what is really relevant to diabetic stem cell mobilopathy is the modulation of cxcl12 in response to mobilization.

- The authors are encouraged to provide a cartoon with the proposed mechanism, give the number of pathways that have been investigated.

Minor

Please consider an overall revision of the manuscript by a native-speaking English reviewer. For example:

- line 189: "distrupt".

- line 305: "exaggerated";

Journal: Communications Biology

Manuscript Title: Dysregulated transforming growth factor-beta mediates early bone marrow dysfunction in diabetes.

Manuscript ID Number: COMMSBIO-22-0960A

Author's response:

We sincerely thank the two reviewers and their insightful comments on our study. As suggested by the reviewers, we have performed additional studies to address the questions raised. In addition, the manuscript has been extensively revised, in part based on the reviewers' suggestions, and to incorporate the new data. We believe that the reviewers' comments and steps we have taken in response have greatly enhanced our study.

Below is our point-to-point response to the comments. Reviewer's comments are in grey shaded text.

REVIEWER 1

Remarks to the Author

This paper reports data regarding the adipogenic features of tibial bone marrow in mice made diabetic using streptozotocin. It also reports the downregulation of TGF β signalling in BM and the effect of high glucose on BM stromal cells to commit to adipogenesis, possibly through a mechanism involving the TGF β signalling pathway.

They report that the adipogenic phenotype is limited to the tibia at the early stage of diabetes before structural changes occur, therefore they conclude that this complication precedes other systemic complications.

The study includes a large number of histological and molecular data, well illustrated by the figures. The experiments seem to be properly conducted. I have the following comments that I hope can help the authors in improving their contribution.

Reviewer 1, comment 1:

I agree that there is a dispute about the best diabetic model. However, the authors seem excessively concerned of this aspect to the extent to dedicate a large part of the introduction on the appropriateness of the model instead of identifying the clinical importance of adipogenesis in the diabetic BM, the possible implications and interaction with other adipogenic sites, and the working hypothesis and objectives.

Response: Thank you very much for this comment. In retrospect, our introduction did not frame the research motivation properly, or articulate the clinical significance of bone marrow adipocytes in diabetes. We have revised the introduction section to include a description of what has been reported in bone marrows of diabetic subjects and models, and evidence that links adipocytes to stem cell deficits in the marrow. In a research publication format, we are unable to include all direct and indirect evidence to link marrow composition changes to possible downstream alterations. However, we believe that we have included some of the important

0960A

linkages. In addition, we have included the rationale and hypothesis. We sincerely hope that this revision properly frames the research study.

Revision(s): Major text revision/new text [lines 68-86; 101-103]. Text on the model system shortened from 18 lines to 11 [lines 88-99]. Text for characterization of the model shortened [lines 127-138].

Reviewer 1, comment 2:

In fact, they simply state that the main scope was to examine the effect of STZ-induced diabetes in C57BL/6 mice, with the goal of identifying early changes that may mediate increased bone marrow adipogenesis. They indeed found the adipogenic accumulation is characterized by excess small adipocytes, but the assumption this is antecedent to complications remains questionable. They used an histological approach but they cannot exclude that functional alterations were already present at the same stage. For instance, it is known that type 1 diabetic children have precocious alterations in left ventricular diastolic function. In a paper published in Diabetes Care, Suys et al. (Diabetes Care 27: 1947–1953, 2004) provided evidence that children and adolescents with type 1 diabetes have altered cardiac structure and function compared with age-matched individuals without diabetes. Subjects included in the study had no cardiac signs or symptoms or diabetes complications and were not taking medications known to modify cardiac structure or function. The most striking findings were in the girls with type 1 diabetes. Similarly, seminal studies showed the exercise proteinuria is present in children with type 1 diabetes (Diabetologia. 1981 Nov;21(5):495-7. doi: 10.1007/BF00257792). Therefore, I suggest the authors to perform functional studies on target organ damage and revisit/discuss these functional data with current evidence of early organ damage.

Response: This is an excellent comment. There is ample experimental data showing that diabetic complications start with early functional derangements that lead to pronounced structural alterations in target tissues and eventually to tissue function failure. In retinopathy, early blood flow changes, primarily mediated by dysregulated endothelin and nitric oxide availability, proceed vascular basement membrane thickening and retinal cell loss. Similarly, prolonged isovolumetric relaxation and impaired early diastolic filling may be the first manifestations of diabetic cardiomyopathy – before severe and easily detectable structural hypertrophy and fibrosis are noted. It is likely that diabetic mice in our study exhibit early functional derangements in some target organs of diabetic complications at 1-month of disease duration. Previous studies in experimental diabetes have shown changes in hemodynamics and glomerular hyperfiltration after 5 weeks of diabetes induction in C57BL/6 mice (PMID: 14600035). In terms of cardiomyopathy, no cardiac functional changes have been observed in C57BL/6 mice after 7 weeks of diabetes onset (PMID: 29402990). Interestingly, diastolic dysfunction was more apparent in STZ-induced diabetic females than males. Similarly, no cardiac functional changes were noted between control and diabetic male FV/N mice after 8 weeks of diabetes onset (PMID: 25737497).

In terms of the bone marrow, however, we currently do not have a ‘characteristic’ functional readout – perhaps mobilopathy could serve this purpose. Since we wished to place bone marrow adipogenesis in the context of well-known structural alterations in other organ systems, we compared histological changes in bone tissues with those in other target organs. All tissue slides

0960A

were read by a Pathologist (co-corresponding on the manuscript). We devoted a significant portion of our results section to outline all histological findings. Prompted by the reviewer's comment, we have examined target organs of diabetic mice at 2-months of disease onset and noted increased fibrosis. This new data has been included in the revised manuscript (Fig. S15). We have also gone through the manuscript carefully so as not to overstate our conclusions regarding the timeline of bone marrow alterations. We sincerely hope that this response is compelling.

Revision(s): New figure [Fig S15] added. Text revision/data cited [lines 267-268].

Reviewer 1, comment 3:

In addition, the clinical relevance of this early phenomenon remains obscure. Can the authors elaborate more the concept? Do they think that the transformation drives those other complications (see our review Bone marrow fat: friend or foe in people with diabetes mellitus? Santopaolo M, Gu Y, Spinetti G, Madeddu P. Clin Sci (Lond). 2020 Apr 30;134(8):1031-1048. doi: 10.1042/CS20200220); or represents an independent phenomenon reflective of a selective susceptibility of BM stromal cells as compared with other cells in different organs?

Response: Thank you for directing us to the extensive review article on bone marrow fat in diabetes. We believe that steps we took to address comment 1 also helps with addressing comment 3. Specifically, revision of the introduction section with a clearer framing of the motivation behind the study helps with the clinical relevance. As the reviewer knows, this is an emerging field in diabetes research and most of the exciting data is indirect. There are, however, important linkages that have been identified between bone marrow abnormalities and subsequent cardiovascular complications. We have highlighted a few of these linkages in the revised manuscript. In terms of bone marrow adipogenesis linking to cardiovascular outcomes in diabetes, data is mostly indirect and associative. We believe that our study takes an important foundational step towards putting the pieces together and link enhanced bone marrow adipogenesis to secondary vascular complications of diabetes.

Revision(s): Minor text revisions in results and discussion section. Major text revision/new text in the introduction section [lines 68-86; 101-103].

Reviewer 1, comment 4:

The study describes a plethora of molecular modifiers, and then focuses on TGF β , which appears to be the second pillar of this paper. Although the data in BM stromal cells seem confirmatory of this implications, I remain concerned about the lack of information regarding the intrinsic mechanism causing the TGF β downregulation. For instance, TGF β could be regulated at transcriptional levels or inactivated through formation of latent complex, that is biologically inactive. How high glucose or lack of insulin interfere with these mechanisms?

Response: The reviewer is absolutely correct that TGF β 1 could be regulated at the posttranslational level, in addition to the transcriptional level. There are no published studies, to our knowledge, that have extensively investigated TGF β 1 expression and regulation in different cell types of bone tissues. Most studies have utilized cell culture models and knocking out TGF β pathway

0960A

genes in mice. Studies in cultured bone cells, however, show that TGFB1 is produced predominantly a latent form lacking the latent TGFB-binding protein (LTBP) (PMID: 8120044). This form is suggested to represent a pool of readily available TGFB1. Although we have not studied TGFB1 latent forms, our data clearly shows that *Tgfb1* mRNA expression is decreased in bone marrows of diabetic mice compared to non-diabetic control mice. Furthermore, the amount of TGFB1 protein in the bone marrow of diabetic mice is decreased. We have performed additional studies to localize the cellular source of TGFB1 by *in situ* hybridization (new figures added: Fig S12 and S13). Our new studies show a robust expression of *Tgfb1* in megakaryocytes (Fig. S12). This is consistent with a recent study showing that megakaryocytes are among the high *Tgfb1* expressers in bone tissues. Other bone marrow cells also expressed *Tgfb1* (Fig. S13). Tissues of diabetic mice showed an overall reduction in *Tgfb1* expression in most bone marrow cells, consistent with our immunohistochemical TGFB1 staining.

The mechanism by which diabetes may reduce TGFB1 signaling may also include downregulation of other signaling factors in the pathway, such as TGFB receptors and SMADs, which we also showed were suppressed in bone tissues of mice. The recent study Daniel Link's group (PMID: 31204302) shows that suppressing TGF signaling increases bone marrow adiposity. Together with our cell culture studies, suppressed TGFB signaling appears to regulate diabetes-induced bone marrow adipogenesis.

We acknowledge that we do not know the latent forms of TGFB1 in bone marrow of diabetic mice and how much TGFB1 may be activated. We have revised the text in our manuscript to highlight this important limitation.

Revision(s): Major text revision/new text [lines 488-500]. New in situ hybridization data figures have been added [Fig. S12 and S13]. Text describing the new data has also been added [lines 250-261].

Reviewer 1, comment 5:

This data on TGFb is also different from our report of enhanced adipogenesis in type 2 diabetic mice (we could not find any change in mRNA levels of TGFb) that the authors did not include in their reference section, but that we think would be relevant to report and discuss/elaborate vs their data (MCP-1 Feedback Loop Between Adipocytes and Mesenchymal Stromal Cells Causes Fat Accumulation and Contributes to Hematopoietic Stem Cell Rarefaction in the Bone Marrow of Patients With Diabetes. Ferland-McCollough D, Maselli D, Spinetti G, Sambataro M, Sullivan N, Blom A, Madeddu P. Diabetes. 2018 Jul;67(7):1380-1394. doi: 10.2337/db18-0044. Epub 2018 Apr 27. PMID: 29703845).

Response: Thank you for suggesting that we include and discuss PMID: 29703845. As we understand it, *Tgfb1* expression was not altered in adipocytes purified from bone marrow of type 2 diabetic subjects when compared to adipocytes from non-diabetic subjects. We have included the study and have briefly discussed it, as suggested by the reviewer.

Revision(s): Major text revision/new text [lines 438-449].

Reviewer 1, comment 6:

0960A

To support their conclusion on TGF β , inducers of TGF β 1 could be used in vivo.

Response: We have been searching for means to induce Tgfb1 in mice. However, we are not aware of any effective Tgfb1 inducers. The only agent that comes to mind is butyrate. A few studies have shown that butyrate increases Tgfb1 in intestinal cells (PMID: 29950699). It is interesting that butyrate, at least in one study, reduced epididymal adipose expansion in ApoE knockout mice fed a high-fat diet (PMID: 29429540). However, it is not clear whether these effects are mediated through Tgfb1 induction or the reported GPCR-mediated mechanism (PMID: 24602606). We also do not know if butyrate would be effective in the bone marrow. In our unpublished studies, we have tested a few Pparg inhibitors which appear to be active/effective in the liver and other tissues, but not bones.

Tamoxifen has also been reported to induce Tgfb1 but there are studies that refute this activity (PMID: 8519657, 12680217).

Revision(s): None.

Reviewer 1, comment 7:

STZ was given according to standard optimized procedure (low repeated doses). Therefore, a possible toxic effect is unlikely responsible for the changes observed in BM. Nonetheless, they found the the animals have a delayed growth. Recent studies have shown that caloric restriction in young, growing mice is associated with impaired skeletal acquisition, low leptin and IGF - 1 levels, and high marrow adiposity (J Bone Miner Res. 2010 Sep; 25(9): 2078–2088. Published online 2010 Mar 12. doi: 10.1002/jbmr.82). How can they exclude that the adipogenic change is related to a dietary defect and/or malabsorption of nutrients in the STZ group? To understand if lack of insulin is the driver of the observed BM changes, rescue experiments with insulin could be very important.

Response: Thank you for this comment. Enhanced bone marrow adiposity has been documented in insulin-deficient type 1 diabetes models – for example, PMID: 17609971 and 16972249 (BALB/c mice with multiple STZ); PMID: 17053023 (NOD mice); PMID: 18162513 (CD-1, multiple STZ). These studies suggest that bone marrow adiposity in diabetes is independent of insulin levels. Presence of insulin, of course, may enhance the process – which may be the case for type 2 diabetes. Studies have shown that insulin may inhibit lipolysis, increase the expression of fatty acid transporters, or modulate various accessory proteins involved in transport of nutrients. It is interesting that some of these accessory proteins may directly be modulated by hyperglycemia. For example, we know that high levels of glucose may modulate mTORC (PMID: 28656269) which is involved in adipogenesis (PMID: 26185979). Studies performed in 3T3-L1 pre-adipocyte cell line suggests that presence of fatty acids is perhaps more important than insulin for adipogenic induction (PMID: 22444967). We also know that STZ induction of diabetes increases free fatty acids as early as 2 weeks following disease onset (PMID: 25395613). Therefore, the likely mechanism of bone marrow adipogenesis involves fatty acid transporters and excess fatty acids. We have conducted a new study in bm-MPCs to show that important fatty acid transporters are expressed, and that cells accumulate intracellular lipids in response to fatty acid exposure. Specially, human bm-MPCs expressed *FATP1*, *ACSL1*,

0960A

and *CD36*. Exposure of these cells to insulin-deficient induction media showed differentiation in the presence of high glucose and/or fatty acids. This new experiment confirms that bone marrow cell adipogenic differentiation is independent of insulin. We have added the new data figure (Fig S10) and revised the text (lines 197-215).

We did not have blood samples from these mice available for leptin and *Igf1* measurements. However, we had access to marrow flush samples and assessed the expression of *Lep* and *Igf1* transcripts. *Lep* transcripts were below the detection level, and *Igf1* levels did not change in marrows of diabetic mice when compared to control mice (Reviewer-only figure below).

Reviewer-only figure: mRNA levels of *Igf1* in the tibia flush samples from control and streptozotocin (STZ)-induced diabetic mice after 1 month of diabetes onset. *Lep* mRNA was also measured in the samples but was not detected [data normalized to *Actb* and *Gapdh*; Mean \pm SD; n = 6; each data point represents a mouse; two-tailed student's t-test: * p<0.05].

Reviewer-only table: qPCR primers for mouse genes. qPCR was performed as described in the manuscript.

Gene	Gene description	Chemistry	Source (Cat#, Reference)
Igf1	Insulin-like growth factor 1	Taqman	Thermo Fisher (Mm00439560_m1)
Lep	Leptin	Taqman	Thermo Fisher (Mm00434759_m1)
Actb	Actin, beta	Taqman	Thermo Fisher (Mm02619580_g1)
Gapdh	Glyceraldehyde-3-phosphate dehydrogenase	Taqman	Thermo Fisher (Mm99999915_g1)

Revision(s): A new data figure [Fig S10] has been added showing the insulin-independent adipogenesis in bm-MPCs. Corresponding text has also been added describing the results [lines 197-215].

Reviewer 1, comment 8:

I also noted that BM of control animals receiving citrate vehicle contains large amount of fat. STZ is prepared in cold citrate buffer at a pH of 4.5 to enhance stability. The authors should consider the possibility that the acidity of the injected citrate had no influence on local pH in BM

0960A

with synergistic effect on STZ induced damage. In fact, we found that even minimal acidosis in the BM can cause osteoclastogenesis and BM remodelling through the Activation of Transient Receptor Potential Cation Channels (10.1038/srep30639).

Response: This is an excellent observation. As the reviewer is aware, different types of bone marrow adipocytes have been noted, dating back to 1976 (PMID: 56163). Recently, MacDougald's group (PMID: 29343445) proposed the presence of *constitutive* bone marrow adipose tissue and *regulated* adipose tissue. Studies have shown that regulated adipose tissue in the marrow changes in response to various conditions, such as exercise (PMID: 28436105), fasting (PMID: 29360620), high-fat diet feeding (PMID: 27512386), and estrogen deficiency (PMIDs: 28824548, 23246792). In rodents, regulated bone marrow adipose tissue is localized to shaft and proximal tibia and femur (PMIDs: 26245716, 24990938). We have carefully provided a schematic next to each analysis, to indicate the region where measurements were taken. We focused on regulated adipose tissue to examine how diabetes enhances marrow adiposity. The adipocytes (please correct us if we have misinterpreted the comment) noted by the reviewer in control tissues are part of the constitutive adipose tissue in distal bone regions (Fig. 2i). We have included this description in the revised manuscript.

Revision(s): Major text revision/new text [lines 179-184].

Reviewer 1, comment 9:

Regarding statistics, I suggest to confirm that the authors used an adjustment for multiple comparisons.

Response: We confirm that we adjusted for multiple comparisons. This confirmation is provided under *Statistics and Reproducibility* section of Supplemental file. In addition, each figure legend indicates the statistical test used.

Revision(s): None.

0960A

REVIEWER 2**Remarks to the Author**

The work of Kum and colleagues explores a relevant topic in diabetology, addressing the remodeling of the bone marrow in diabetes.

The authors studied the effects of hyperglycemia on the remodeling of the bone marrow in terms of enhance adipogenesis. These findings are not entirely new, as it has been shown that type 1 diabetes is associated with increased adiposity in the bone marrow (PMID: 32345753). The authors should refer to this work when commenting their findings.

The work has some merit, though. In particular the quality of histological images is excellent. However, some flaws and major issued must be addressed.

Reviewer 2, Major Comments**Reviewer 2, comment 1:**

The authors shall greatly condense the first paragraph of the "Results". Indeed, it is well established that the streptozotocin-induced diabetic mice do not develop some complications unless after > 2 month of hyperglycemia. Furthermore, other groups have already showed that 1 month of hyperglycemia is sufficient to induce robust alterations in the bone marrow (PMID: 32345753, 23663738, 32316750).

Response: Thank you for this suggestion. We included a full characterization of the model because different researchers induce diabetes in mice at different starting age – for example, from 6 weeks of age to 12 weeks of age. At times, age at diabetes induction and follow-up period is not even clearly specified. It is true that for some assays, it may not make a difference. We do wish, however, that most researchers provided a full characterization with links to established structural/functional benchmarks. This allows researchers/readers to place their observations in the proper context.

As suggested by the reviewer, we have shortened the first paragraph of our Results section [lines 123-146 shortened to 127-138]. In addition, we have shortened the rationale for the model in our introduction [lines 78-96 shortened to 88-99].

Revision(s): Text revised [lines 127-138; 88-99].

Reviewer 2, comment 2:

The authors analyzed epididymal fat and reported an increased number of adipocytes, which is suggestive of hyperplasia. However, this is highly unlikely in the setting of this model of diabetes. Indeed, streptozotocin-induced diabetes induces a massive lipodystrophic phenotype due to the lack of insulin and a profound atrophy of the adipose tissue. This is compatible with the gene expression profile where all the major adipose tissue related genes are downregulated. The increased number of adipocytes is therefore the result of lipolysis due to counterregulatory mechanism activated by the lack of insulin. The authors should revise their conclusion from this set of data.

0960A

Response: This is an excellent point. We have revised our conclusions from this set of data.

Revision(s): Text revised [lines 160-161].

Reviewer 2, comment 3:

Line 208. The authors claim that diabetes does not alter hematopoietic areas, confirming "previous reports". However, diabetes is known to modify hematopoiesis triggering myelopoiesis. Furthermore, diabetes triggers neuropathy and capillary rarefaction (PMID: 24270983, 26358583, 20042708). Thus, the authors should better explain this sentence.

Response: We sincerely apologize for the vague description. We have revised the text to clearly indicate what was reported in the two studies cited. Specifically, both studies measured the number of CD45-positive cells in the bone marrow. In our current study, we noted the appearance of adipocytes prior to a reduction in CD45 signal. In PMID 20042708, researchers have shown a reduced hematopoietic fraction in mice after 27-30 week of diabetes onset. Together, these results show the bone marrow adiposity may proceed hematopoietic fraction alteration as determined by (immuno)histological/morphometric analyses.

Revision(s): Text revised [lines 219-225].

Reviewer 2, comment 3:

Line 2016. While, Sca1 is a stemness marker for HSC, it is also expressed by stromal cells. A flow cytometry approach to quantify the stem cell compartment is more suited for such quantification. Furthermore, diabetes has been associated with some changes in the HSC compartment in previous works. The authors should comment about that.

Response: Thank you for this comment. As suggested, we have revised the text to acknowledge that SCA1 is not specific to hematopoietic stem cells. This was one of the reasons that we included an additional (more selective) marker of stem cells: SOX2. In addition, we measured transcript levels of four stem cell-associated genes: *Sca1*, *Sox2*, *Oct4*, and *Nanog*. Of these transcripts, *Nanog* and *Sox2* are among the most stem cell restricted. Lastly, in the bone marrow, *Cxcr4* is also present on stem cells and was measured in our study. We have added the comment as suggested by the reviewer.

Revision(s): Text revised [lines 232-234].

Reviewer 2, comment 4:

In which cells of the bone marrow the TGF-B signaling is reduced in diabetes in vivo? Mesenchymal stromal cells? The authors should show some data about this point in vivo, using immunofluorescence or cell sorting.

Response: This is an excellent question. We included immunofluorescence images of bone tissues stained for TGFB1 (Fig. 3j). Our data shows TGFB1 immunoreactivity around most bone marrow cells in tissues harvested from control mice. Examination of tissues from diabetic mice showed widespread reduction in TGFB1 immunoreactivity. In an attempt to localize the

0960A

cellular source of TGFB1, we performed in situ hybridization (new figures added: Fig S12 and S13). Our studies show a robust expression of *Tgfb1* in megakaryocytes (Fig. S12). This is consistent with a recent study showing that megakaryocytes are among the high *Tgfb1* expressers in bone tissues. Other bone marrow cells also expressed *Tgfb1* (Fig. S13). Tissues of diabetic mice showed an overall reduction in *Tgfb1* expression in most bone marrow cells – consistent with our immunohistochemical TGFB1 staining.

Revision(s): New in situ hybridization data figures have been added [Fig. S12 and S13]. Text describing the new data has also been added [lines 250-261].

Reviewer 2, comment 5:

Line 256: The authors should check also for other niche factors like Vcam and Angpt1.

Response: Thank you for this suggestion. We measured the suggested niche factors: Angpt1, Icam1, and Vcam1. Our data shows no significant changes in the expression of these niche factors in the bone marrow of diabetic mice. We have added the new data figure in the revised manuscript.

Revision(s): A new data figure [Fig. S17] has been added. Corresponding text has also been added [lines 287-288].

Reviewer 2, comment 6:

The authors should consider that adipocyte in the bone marrow have higher expression of Cxcl12 compared to other cells (PMID: 32345753). Therefore, the actual amount of Cxcl12 in the tibiae of diabetic mice might be higher than non-diabetic mice. Indeed, when you flush the bone marrow by centrifugation, and you resuspend the pellet, you'll inevitably lose the adipocytes, unless you collect them separately or lyse the pellet without resuspending it. The authors should consider that when they comment their data.

Response: Thank you. As suggested, we have considered the possibility that adipocytes (an important source of CXCL12) may have been removed when preparing our marrow flush samples for gene expression studies. However, it is important to note that we have also assessed intact bone tissues for CXCL12 protein expression (Fig. S16i). Furthermore, femur tissues, which do not contain significant adipocytes in control and diabetic mice, also showed a reduction in *Cxcl12* expression (Fig S16a, S16k). Taken together, our data shows a reduction in *Cxcl12* expression in tibia and femur of diabetic mice. We have revised the text to inform readers that an important cellular course of CXCL12 may have been omitted during sample preparation.

Revision(s): New text included [lines 452-460].

Reviewer 2, comment 7:

Line 436. The authors did not study mobilization so the claim about the role of their data in the mechanism of diabetic stem cell mobilopathy is not supported. Despite the reduced level of *Cxcl12* in the bone marrow, what is really relevant to diabetic stem cell mobilopathy is the modulation of *cxcl12* in response to mobilization.

0960A

Response: We agree with the reviewer. We did not study mobilization and have therefore, revised the text. It is absolutely true that a reduction in Cxcl12 following exposure to a mobilizing agent is not seen in diabetic mice. However, the study also noted that diabetic mice have significantly reduced overall expression of Cxcl12 (PMID: 21998408). As the reviewer implied, mobilization defects relate to disruption of a Cxcl12 gradient. It is possible that reduced overall levels of Cxcl12 in diabetes may participate in this disrupted gradient. Text has been revised to remove unsupported or overstated conclusions.

Revision(s): Text revised [lines 463-472].

Reviewer 2, comment 8:

The authors are encouraged to provide a cartoon with the proposed mechanism, give the number of pathways that have been investigated.

Response: Thank you for this suggestion. We have included a schematic of the major findings and proposed mechanisms (Fig. 8).

Revision(s): We have added a new figure [Figure 8]. Figure is cited in the first paragraph of the discussion section [line 421-422].

Review 2, Minor Comments

Reviewer 2, comment 1:

Please consider an overall revision of the manuscript by a native-speaking English reviewer. For example:

- line 189: "distrupt".
- line 305: "exaggerated";

Response: We regret that we missed a few spelling and grammatical errors. The manuscript was drafted and reviewed by native English speakers prior to submission. We have proofread the revised manuscript to ensure there are no remaining spelling and grammatical errors.

Revision(s): Minor corrections made throughout the manuscript.

—

REVIEWERS' COMMENTS:

Reviewer #1 (Remarks to the Author):

The present version is improved and responds to my previous questions.

Reviewer #2 (Remarks to the Author):

The authors provided detailed argumentations to the issue raised and provided additional data to support their claims.

As such, the work is now acceptable for publications.